# STRUCTURE-GUIDED LARGE LANGUAGE MODELS FOR TEXT-TO-SQL GENERATION

## ABSTRACT

Recent advancements in large language models (LLMs) have shown promise in bridging the gap between natural language queries and database management systems, enabling users to interact with databases without the background of SQL. However, LLMs often struggle to fully exploit and comprehend the user intention and complex structures of databases. Decomposition-based methods have been proposed to enhance the performance of LLMs on complex tasks, but decomposing SQL generation into subtasks is non-trivial due to the declarative structure of SQL syntax and the intricate connections between query concepts and database elements. In this paper, we propose a novel **Structure GUided text-to-SQL framework (`SGU-SQL`)** that incorporates syntax-based prompting to enhance the SQL generation capabilities of LLMs. Specifically, `SGU-SQL` establishes structure-aware links between user queries and database schema and recursively decomposes the complex generation task using syntax-based prompting to guide LLMs in incrementally constructing target SQLs. Extensive experiments on two benchmark datasets demonstrate that `SGU-SQL` consistently outperforms state-of-the-art text-to-SQL baselines. These results highlight the importance of incorporating structural syntax information for effective text-to-SQL generation and pave the way for more robust and reliable interfaces to databases in the era of artificial intelligence.

## 1 INTRODUCTION

Text-to-SQL is a challenging task that aims to bridge the gap between natural language queries and database management systems, enabling users to interact with databases without knowing the background of SQL. In the past few years, this task has been incrementally evolving due to the complexity of SQL syntax and the intricate connections between user queries and database elements. Models need to interpret intricate natural language queries and construct SQL queries with precise syntax structure, all while linking with correct tables and columns in the database. A wide range of research has been proposed to address these issues, including intermediate query languages, graph-based modeling, and skeleton query generation (Wang et al., 2019; Li et al., 2023a;b).

Recently, this field has seen significant progress with the emergence of Large Language Models (LLMs) like GPT series (Radford et al., 2018; Achiam et al., 2023; Brown et al., 2020; OpenAI, 2023). Training on a wide array of corpus, LLMs exhibit exceptional ability in understanding and producing text that closely mimics human communication. Researchers have started exploring the potential of LLMs for text-to-SQL by leveraging their extensive knowledge reserves and superior generation capabilities (Rajkumar et al., 2022; Gao et al., 2024). These approaches often involve prompt engineering to guide proprietary LLMs in SQL generation (Chang & Fosler-Lussier, 2023; Pourreza & Rafiei, 2023) or fine-tuning open-source LLMs on text-to-SQL datasets (Gao et al., 2024).

Despite their advancements, LLM-based text-to-SQL models encounter several limitations that impede their successful application in practice.

❶ **Ambiguous User Intent.** Accurately interpreting the user's intent in natural language remains a significant challenge for LLM-based models. Natural language is inherently ambiguous and context-dependent, making it difficult for models to discern precise requirements. For example, a query like "Show me last quarter's sales performance" requires the model to infer specific details such as relevant tables, metrics defining "performance" and the exact time frame for "last quarter".

Additionally, nuanced language involving implied conditions or comparisons, such as `better than average` or `most recent` can lead to misinterpretations, resulting in queries that do not fully align with the user's intent.

❷ **Sophisticated Database Architecture.** Mapping natural language terms to specific database columns and tables is another critical area where LLM-based models struggle. Databases often have complex schemas with interrelated tables and non-intuitive naming conventions. For instance, a user referring to `customer purchases` might imply multiple tables like `Customers`, `Orders` and `OrderDetails` The model must accurately identify and relate these tables, which is challenging without comprehensive schema awareness. Moreover, similar column names across different tables can cause confusion, leading to incorrect selections and incomplete queries, especially in large or poorly documented databases.

❸ **Complex Syntax Structure of SQL.** Generating syntactically accurate and logically coherent SQL queries is a challenging task. SQL requires precise clause arrangement, correct operator usage, and adherence to grammatical rules. LLMs may produce queries with syntax errors, such as missing commas, incorrect JOIN conditions, or misplaced keywords. Constructing complex queries involving nested subqueries, aggregate functions, or window operations demands high precision, which is typically beyond the current capabilities of LLMs. Recently, decomposition-based methods have been proposed to enhance the performance of LLMs on complex tasks. However, decomposing the complicated linked structure into smaller, manageable components for step-by-step SQL generation requires effective strategies. Traditional approaches often struggle with handling complex queries due to the declarative structure of SQL and the intricate connections between user queries and database elements.

In this paper, we propose a novel Structure Guided text-to- SQL framework (`SGU-SQL`). `SGU-SQL` addresses the above issues by leveraging the structural information in queries and databases through structure-aware linking and syntax-based decomposition, providing additional guidance to the LLM for better SQL generation. Specifically, `SGU-SQL` represents user queries and databases into unified and structured graphs and employs a tailored structure-learning model to establish a connection between the user queries and the databases. The linked structure is then decomposed into sub-syntax trees, guiding the LLMs to generate the SQL query incrementally. Our main contributions are summarized as follows:

- We identify the limitations of LLM-based Text-to-SQL models and introduce `SGU-SQL`, which leverages structural syntax information to improve SQL generation capabilities of LLMs.
- `SGU-SQL` proposes graph-based structure construction to comprehend user query and database structure and then link query and database structure with dual-graph encoding.
- `SGU-SQL` introduces tailored structure-decomposed generation strategies to decompose queries with syntax trees and then incrementally generate accurate SQL with LLM.
- Experiments on two benchmarks verify that `SGU-SQL` outperforms state-of-the-art baselines, including 11 fine-tuning models, 7 structure learning models, and 14 in-context learning models.

## 2 PROBLEM STATEMENT

Let $\mathcal{D}$ be a database schema consisting of a set of tables $\mathcal{T} = \{T_1, T_2, \ldots, T_n\}$, where each table $T_i$ has a set of columns $\mathcal{C}_i = \{C_{i1}, C_{i2}, \ldots, C_{im}\}$. The database schema $\mathcal{D}$ can be represented as a tuple $(\mathcal{T}, \mathcal{C})$, where $\mathcal{C} = \bigcup_{i=1}^{n} \mathcal{C}_i$. Using the above notations, we describe our problem below.

**Definition 1. Structure Learning for Text-to-SQL**: Given a natural language query $\mathcal{D}$ and a database schema $\mathcal{Q}$, the task of graph learning for Text-to-SQL aims to generate a graph-based representation $\mathcal{G}$ that captures the structural and semantic relationships between the query and the schema, and to learn a mapping function $f : \mathcal{G}_q \to \mathcal{G}_d$, where $\mathcal{G}_q$ is the structural user queries, and $\mathcal{G}_d$ is the corresponding database contents linked to the query $\mathcal{G}_q$.

**Definition 2. Text-to-SQL Generation**: Given a natural language query $Q$ and a database schema $\mathcal{D}$, the task of Text-to-SQL generation aims to translate $Q$ into a corresponding SQL query $S$ that accurately retrieves the desired information from the database.

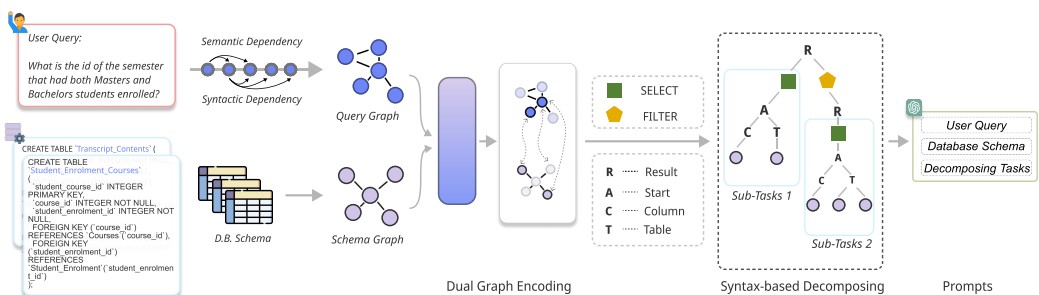

Figure 1: The overall framework of SGU-SQL.

# 3 THE FRAMEWORK OF SGU-SQL

In this section, we will introduce the key components of SGU-SQL in detail. We leverage the implicit structural information in both queries and databases from three aspects: $(i)$ A graph-based structure construction for both user query and database understanding; $(ii)$ A tailored structure linking method is proposed to map the natural language query to the relevant database elements. $(iii)$ Structure-based prompting to LLMs for accurate SQL generation, which decomposes the complex generation into sub-tasks and guides LLMs to generate the SQL query incrementally, adhering to the necessary syntax structure. The overall illustration is presented in Figure 1.

## 3.1 REVISITING USER QUERY AND DATABASE VIA GRAPH

Bridging the gap between textual queries and the structured database poses several challenges. Firstly, constructing an accurate structure that captures the relationships between query terms and database entities is a non-trivial task. Secondly, linking the query to the appropriate tables and columns in the database is challenging, especially when there is ambiguity or a lack of explicit connections. In this paper, we build a comprehensive query-schema graph designed to structure the query concept, the schema, and pre-defined relations between the query phrases and the tables or columns present within the schema. The graph contains three key structures: $(i)$ **Query Structure** $(R_q)$: Encodes dependencies between tokens in the question, derived from its syntactic parse. $(ii)$ **Database Structure** $(R_s)$: Represents intrinsic relationships within the database schema, like foreign keys. $(iii)$ **Linking Structure** $(R_l)$: Aligns query entities with the columns or tables in the database.

### 3.1.1 USER QUERY UNDERSTANDING AND REPRESENTATION

A query graph can be depicted as $\mathcal{G}_q = (V_q, R_q)$, where $V_q$ denotes the node set that characterizes the keywords specified in the question, and $R_q$ signifies the relationships among these keywords. To differentiate the relationship between various words, we establish three separate link categories, including `Forward-Syntax`, `Backward-Syntax` and `None-Syntax` relations as defined in Table 9, to encapsulate the particular syntactic connections among words in the vernacular question.

a) Query Parsing: Syntactic parsing can help resolve structural ambiguities in the query by providing a hierarchical representation of the sentence structure. Specifically, we first define a context-free grammar $G_q$ for the query language:

$$G_q = (N_q, \Sigma_q, P_q, S_q), \tag{1}$$

where $N_q$ is a finite set of non-terminal symbols representing query concepts. $\Sigma_q$ is a finite set of terminal symbols representing query terms. $P_q$ is a finite set of production rules that map non-terminals to sequences of terminals and non-terminals. $S_q \in N_q$ is the start symbol.

The production rules $P_q$ define the syntactic structure of the query language. For example, the set of production rules of SQL is listed in Figure 2-(a) of the Appendix.

Parsing a user query $Q$ using the grammar $G_q$ yields a syntax tree $T_q = (V_q, E_q)$, where $V_q$ is the set of vertices representing query concepts. $E_q \in R_q$ is the set of edges representing syntactic relationships between the query concepts.

b) Coreference resolution: Natural language queries often contain ambiguities, such as polysemy (words with multiple meanings) and syntactic ambiguity (multiple possible syntax trees). Let $Q$ be the set of all possible interpretations of a query $q$. The ambiguity challenge can be formulated as selecting the most likely interpretation $\hat{q}$ from $Q$:

$$\hat{q} = \arg\max_{q_i \in Q} P(q_i \mid q), \tag{2}$$

where $P(q_i \mid q)$ is the probability of interpretation $q_i$ given the original query $q$.

Natural language queries may contain multiple mentions of the same entity, which need to be resolved to construct an accurate graph representation. Let $M$ be the set of entities mentioned in the query and $E$ be the set of unique entities. The coreference resolution can be formulated as finding a mapping function $\phi : M \to E$ that maps each mention to its corresponding entity:

$$\phi(m) = \text{argmax}_{e \in E} P(e \mid m), \tag{3}$$

where $P(e \mid m)$ is the probability of entity $e$ given the mentioned entity $m$.

c) Query Graph Construction: Once the syntax tree $T_q$ is obtained, we can construct the graph structure $\mathcal{G}_q = (V_q, E_q)$ representing the user query. The vertices $V_q = V_q$ is the set of query concepts and terms and edges $E_q$ are defined as follows:

$$E_q = E_q \cup (v_i, v_j) \mid v_i, v_j \in V_q \wedge \text{relation}(v_i, v_j). \tag{4}$$

The edges $E_q$ in the graph structure include both the syntactic relationships from the syntax tree and additional edges based on semantic relationships between query concepts/terms as decided in Table 9. The resulting graph structure $\mathcal{G}_q$ captures both the syntactic structure of the user query and the semantic relationships between query concepts/terms.

### 3.1.2 DATABASE UNDERSTANDING AND REPRESENTATION

To generate accurate SQL queries, text-to-SQL systems also need to have a comprehensive understanding of the database structure, including table names, column names, and relationships between or across various tables/columns. Representing and encoding the database in a way that can be effectively utilized by the text-to-SQL model is a challenging task. In this paper, we introduce a schema graph to represent database structure. Specifically, let $\mathcal{D}$ be a database consisting of a set of tables $\mathcal{T} = T_1, T_2, \ldots, T_n$. Each table $T_i \in \mathcal{T}$ has a set of columns $\mathcal{C}_i = \{C_{i1}, C_{i2}, \ldots, C_{im}\}$. We define a database schema graph $\mathcal{G}_d = (V_d, R_d)$ to represent the structure of the database schema, where $S$ denotes the set of nodes representing tables and columns, and $R_d$ is the set of edges representing relationships between them.

a) Node Representation: Each table $T_i \in \mathcal{T}$ is represented as a node $v_{T_i} \in S$ in the schema graph. Similarly, each column $C_{ij} \in \mathcal{C}i$ of table $T_i$ is represented as a node $vC_{ij} \in S$. The set of nodes $S$ in the schema graph is defined as:

$$S = v_{T_i} \mid T_i \in \mathcal{T} \cup v_{C_{ij}} \mid C_{ij} \in \mathcal{C}_i, T_i \in \mathcal{T}. \tag{5}$$

b) Edge Representation: The relationships between tables and columns in the database schema are represented as edges in the schema graph. As shown in Table 9, we define the following three types of edges:

- Table-Column Edges: For each column $C_{ij} \in \mathcal{C}_i$ of table $T_i$, we add an edge $E\{T_i, C_{ij}\} \in R_S$ connecting the table node $v_{T_i}$ to the column node $v_{C_{ij}}$. This edge represents the relationship between a table and its columns.

$$E(T_i, C_{ij}) = \{v_{T_i}, v_{C_{ij}}, \text{"has"}\}. \tag{6}$$

- Primary-Key Edges: If a column $C_{ij} \in \mathcal{C}_i$ is the primary key column of table $T_i$, we add an edge $E\{C_{ij}, T_i\} \in R_d$ connecting the corresponding column nodes $v_{C_{ij}}$ and the table $v_{T_i}$. The primary-key relations in the schema graph provide information about the structure and integrity constraints of the database.

$$E(T_i, C_{ij}) = \{v_{T_i}, v_{C_{ij}}, \text{"primary\_key"}\}. \tag{7}$$

- Foreign-Key Edges: If a column $C_{ij} \in \mathcal{C}_i$ of table $T_i$ is a foreign key referencing a primary key column $C_{kl} \in \mathcal{C}_k$ of table $T_k$, we add an edge $E\{C_{ij}, C_{kl}\} \in R_d$ connecting the corresponding column nodes $v_{C_{ij}}$ and $v_{C_{kl}}$. This edge represents the foreign key relationship between the columns.

$$E(C_{ij}, C_{kl}) = \{v_{C_{ij}}, v_{C_{kl}}, \text{"foreign\_key"}\}. \tag{8}$$

### 3.1.3 STRUCTURE LINKING WITH DUAL GRAPH ENCODING

The syntax tree $T_q$ obtained from parsing the user query $Q$ captures the syntactic structure of the query. It represents the hierarchical relationships between query concepts and terms, which is crucial for understanding the intent behind the query. By incorporating the syntax tree into the query graph $G_q$, we preserve the syntactic structure of the query and its inherent meaning. The schema graph $G_d$ represents the structure of the database schema, with vertices representing tables and columns and edges representing their relationships. By combining the syntax tree with the schema graph through the mapping function $\phi$, we establish a link between the query concepts/terms and the corresponding schema elements. This mapping allows us to identify which tables and columns in the database are relevant to the user query, enabling more accurate and targeted querying.

Specifically, given the constructed query and database graphs, we value the adjacency information during the matching process and propose to automatically build the connection between the query structure and schema at the node level. Specifically, we design a tailored structure-based linking framework. Both query and schema structures are first encoded through a Relational Graph Attention Network (RGAT) (Busbridge et al., 2019) for initial node representations. The representation learning process is guided by the message propagation within the self-structure. We formalize the procedure of structure-aware question-schema structure linking as follows:

$$\mathcal{G}'_d = \text{Agg}(\mathcal{G}_d, \mathcal{G}_q), \tag{9}$$

$$\mathcal{G}'_q = \text{Agg}(\mathcal{G}_q, \mathcal{G}_d), \tag{10}$$

where the structure-aware aggregation function $\text{Agg}(.)$ is employed to gather information from both the schema-graph $\mathcal{G}_d$ and the query-graph $\mathcal{G}_q$ and transfer it to the adjacent graph.

Let $\{\boldsymbol{h}_i^q\}_{i=1}^m$ represent a set of node embeddings in the query graph $\mathcal{G}_q$ and let $\{\boldsymbol{h}_j^k\}_{j=1}^n$ denote a set of node embeddings in the subgraph $\mathcal{G}_k$ that extracted from the schema graph $\mathcal{G}_d$. In particular, we first employ global-average pooling on the node embedding $\boldsymbol{h}_i^q$ of the query structure $\mathcal{G}_q$ to derive the global query structure embedding $\boldsymbol{h}_g^q$. Following this, to encapsulate globally pertinent information, the key node embedding $\boldsymbol{h}_j^k$ is updated subsequently:

$$\boldsymbol{h}_g^q = \frac{1}{m} \sum\nolimits_{i=1}^m \boldsymbol{h}_i^q, \tag{11}$$

$$\alpha_j = \theta \left( \boldsymbol{h}_g^{qT} \boldsymbol{W}_g \boldsymbol{h}_j^k \right), \tag{12}$$

$$\boldsymbol{h}_j^k = \sum\nolimits_{l \in \mathcal{N}_j} \alpha_l \boldsymbol{W}_k \boldsymbol{h}_l^k + \alpha_j \boldsymbol{W}_k \boldsymbol{h}_j^k \tag{13}$$

$$+ (1 - \alpha_j) \boldsymbol{W}_q \boldsymbol{h}_g^q \tag{14}$$

where $\boldsymbol{W}_g$, $\boldsymbol{W}_q$, $\boldsymbol{W}_k$ represent trainable parameters, and $\theta$ illustrates a sigmoid function. While $\alpha_j$ denotes the relevance score situated between the $j$-th key node and the global query structure.

For each node $a$ in the query structure $\mathcal{G}_q$, it is necessary to find a corresponding matching node $s$ in the database $\mathcal{G}_d$. The proposed solution mainly consists of three steps. First, a set of most relevant candidate nodes $\{s_1, s_2, \ldots, s_K\}$ is identified through string matching in the set of tables and columns $V$. Second, for each candidate node $s$, an enclosing subgraph $\mathcal{G}(a, s)$ is constructed. As shown in Figure 1, $\mathcal{G}(a, s)$ includes the query graph $\mathcal{G}_q$, adjacent nodes of $s_k$, and an edge connecting $a$ and $s_k$. Lastly, we adopt a structure learning model $\text{RGAT}(\cdot)$ to learn the graph-level representation of $\mathcal{G}_{(a, s_k)}$ that captures the compatibility between natural language concepts and database elements.

$$\boldsymbol{h} = \text{RGAT}(\mathcal{G}(a, s)). \tag{15}$$

The matching score of the candidate pair $(a, s_k)$ is then measured by the degree of compatibility:

$$\text{Score}_{(a, s_k)} = \sigma(\sum\nolimits_{l \in \mathcal{G}(a, s_k)} \boldsymbol{h}_l^k). \tag{16}$$

Based on positive samples $(a, s)$ and negative samples $(a, s_k)$, where $s_k \neq s$, the structure learning model $\text{RGAT}(\cdot)$ is iteratively trained:

$$\mathcal{L}_{\text{RGAT}} = -\min \sum_{a_i \in \mathcal{G}_q} \log \frac{\exp(\text{Score}(a_i, s))}{\exp(\text{Score}(a_i, s)) + \sum_{s \in \mathcal{G}_s, s_k \neq s} \exp(\text{Score}(a_i, s_k))}. \tag{17}$$

This contrastive training objective encourages the model to maximize scores for correct matches while minimizing scores for incorrect ones. Through this process, the matching scores evolve into reliable indicators that guide the selection of correct database elements during SQL generation.

**Incorporating pre-defined relations**: After we got the accurate linking from the structure learning model, we further incorporated several additional relations to supplement effective connections between the user query and database schema, which is defined in Table 9. The mapping function $\phi$ relies on a set of pre-defined relations $R$ between query concepts/terms and schema elements. These relations capture the semantic connections between the query and the database schema. By incorporating these relations into the query-schema graph construction, we ensure that the final graph not only captures the syntactic structure of the query but also incorporates the semantic relationships between the query and the schema.

### 3.2 STRUCTURE-DECOMPOSED PROMPTING WITH SYNTAX TREE

#### 3.2.1 DECOMPOSING QUERY WITH SYNTAX TREE

The performance of LLMs on complex tasks can be improved by using decomposing-based methods. However, decomposing a SQL query into subtasks is challenging due to its declarative structure and the intricate connections between query concepts. To this end, in this section, we introduce a context-free syntax tree that defined in Figure 2-(a) to break down the text-to-SQL generation task into smaller subtasks according to the syntax structure of the user query. Specifically, we first employ the query parsing described in Section 3.1.1 to build the syntax tree to achieve a linguistic understanding of the natural language query and then adopt a node mapper to match nodes in the linguistic syntax tree to SQL operations (Kate, 2008). Following this, the original query can be divided into several subtasks according to the SQL operations distributed on the syntax tree.

#### 3.2.2 SUBTASK DECOMPOSITION

Given the context-free syntax tree $\mathcal{T}$, we decompose the generation task into subtasks based on the syntactic structure of the query. Each non-terminal node $n \in N$ in the tree represents a subtask that needs to be solved to generate the corresponding part of the SQL query. The decomposition process $f : N \to S$ that maps each non-terminal node to its corresponding SQL component, is illustrated at Algorithm 1.

#### 3.2.3 SQL GENERATION

To generate the SQL component $s_n$ for a non-terminal node $n \in N$, we employ a LLM $\mathcal{M}$ that takes the natural language query $Q$ and the subtask context $c_n$ as input and produces the corresponding SQL component:

$$s_n = \mathcal{M}(Q, c_n). \tag{18}$$

The subtask context $c_n$ captures the relevant information from the context-free syntax tree $\mathcal{T}$ that is needed to generate the SQL component for node $n$. It can include the parent node, sibling nodes, and other relevant contextual information. The final SQL query $S$ is obtained by combining the SQL components generated for all the non-terminal nodes in the context-free syntax tree $\mathcal{T}$, starting from the root node $n_0$: $S = s_{n_0}$. By decomposing the text-to-SQL generation task into subtasks based on the syntax structure of the user query, we can leverage the hierarchical information captured by the context-free syntax tree to generate more accurate and structured SQL queries.

### 4 EXPERIMENTS

This section empirically evaluates the proposed SGU-SQL, and presents its performance on two benchmark datasets. Our empirical study is motivated by the following questions: **Q1** How does

our proposed SGU-SQL perform in comparison with the strongest baselines, including traditional finetuning-based, structure-learning-based methods, and other in-context-learning-based methods? **Q2** Could our proposed Gram enhance other LLMs by substituting the original framework with the structure-decomposing-based prompt? **Q3** Is our proposed structure prompting effective when handling queries of different complexity? **Q4** Which type of queries are prone to errors in our model? And what is the reason for the error?

## 4.1 EXPERIMENT SETUP

| Text-to-SQL Method | Backbone LM/LLM | Finetuning | Structure Information | Prompt Strategy | SPIDER | | | | |
|---|---|---|---|---|---|---|---|---|---|
| | | | | | Easy | Medium | Hard | Extra | Overall |
| Baichuan2 | Baichuan2-7B | SFT | ✘ | ✘ | 0.5775±0.0106 | 0.3521±0.0130 | 0.2010±0.0089 | 0.0667±0.0115 | 0.3353±0.0125 |
| | | LoRA | ✘ | ✘ | 0.8714±0.0073 | 0.6305±0.0069 | 0.4489±0.0063 | 0.2958±0.0084 | 0.6035±0.0079 |
| | | QLoRA | ✘ | ✘ | 0.8919±0.0057 | 0.6367±0.0071 | 0.4885±0.0053 | 0.3306±0.0079 | 0.6242±0.0061 |
| | Baichuan2-13B | SFT | ✘ | ✘ | 0.5805±0.0093 | 0.4133±0.0085 | 0.2644±0.0067 | 0.1875±0.0078 | 0.3927±0.0081 |
| | | LoRA | ✘ | ✘ | 0.9024±0.0075 | 0.7015±0.0069 | 0.5688±0.0083 | 0.3915±0.0071 | 0.6776±0.0080 |
| | | QLoRA | ✘ | ✘ | 0.8951±0.0103 | 0.6746±0.0123 | 0.5809±0.0115 | 0.3434±0.0109 | 0.6592±0.0114 |
| LlaMA2 | LlaMA2-7B | LoRA | ✘ | ✘ | 0.8868±0.0016 | 0.6410±0.0041 | 0.4892±0.0030 | 0.3311±0.0017 | 0.6259±0.0022 |
| | | QLoRA | ✘ | ✘ | 0.8472±0.0025 | 0.6234±0.0032 | 0.4658±0.0021 | 0.3309±0.0027 | 0.6083±0.0035 |
| | LlaMA2-13B | LoRA | ✘ | ✘ | 0.9066±0.0037 | 0.7292±0.0045 | 0.5517±0.0029 | 0.3430±0.0055 | 0.6809±0.0030 |
| | | QLoRA | ✘ | ✘ | 0.9110±0.0043 | 0.7004±0.0059 | 0.5523±0.0032 | 0.3190±0.0061 | 0.6648±0.0045 |
| | LlaMA2-70B | SFT | ✘ | ✘ | 0.4110±0.0093 | 0.2293±0.0075 | 0.1906±0.0081 | 0.0725±0.0090 | 0.2414±0.0108 |
| | | LoRA | ✘ | ✘ | 0.9151±0.0069 | 0.7323±0.0080 | 0.5575±0.0049 | 0.3921±0.0035 | 0.6869±0.0040 |
| CodeLlama | CodeLlama-7B | SFT | ✘ | ✘ | 0.2136±0.0150 | 0.1769±0.0161 | 0.0921±0.0169 | 0.0363±0.0144 | 0.1487±0.0163 |
| | | LoRA | ✘ | ✘ | 0.9228±0.0105 | 0.7562±0.0134 | 0.5863±0.0096 | 0.3485±0.0126 | 0.7018±0.0108 |
| | | QLoRA | ✘ | ✘ | 0.9115±0.0127 | 0.7506±0.0142 | 0.5982±0.0120 | 0.3310±0.0085 | 0.6961±0.0104 |
| | CodeLlama-13B | SFT | ✘ | ✘ | 0.6980±0.0115 | 0.6015±0.0121 | 0.4073±0.0109 | 0.2708±0.0145 | 0.5288±0.0140 |
| | | LoRA | ✘ | ✘ | 0.9414±0.0086 | 0.7885±0.0073 | 0.6842±0.0081 | 0.4041±0.0069 | 0.7462±0.0092 |
| | | QLoRA | ✘ | ✘ | 0.9402±0.0053 | 0.7445±0.0066 | 0.6263±0.0085 | 0.3915±0.0061 | 0.7270±0.0085 |
| | CodeLlama-70B | SFT | ✘ | ✘ | 0.7223±0.0143 | 0.6245±0.0120 | 0.4432±0.0131 | 0.3028±0.0147 | 0.5675±0.0144 |
| | | LoRA | ✘ | ✘ | **0.9621±0.0053** | 0.8122±0.0069 | 0.7167±0.0055 | 0.4324±0.0069 | 0.7710±0.0061 |
| Qwen | Qwen-7B | SFT | ✘ | ✘ | 0.3956±0.0155 | 0.2561±0.0131 | 0.1384±0.0137 | 0.0427±0.0169 | 0.2356±0.0140 |
| | | LoRA | ✘ | ✘ | 0.8546±0.0060 | 0.6876±0.0089 | 0.5743±0.0076 | 0.3340±0.0065 | 0.6519±0.0073 |
| | | QLoRA | ✘ | ✘ | 0.9110±0.0045 | 0.6747±0.0081 | 0.5750±0.0076 | 0.3436±0.0055 | 0.6623±0.0069 |
| | Qwen-14B | SFT | ✘ | ✘ | 0.8713±0.0105 | 0.6323±0.0140 | 0.3686±0.0139 | 0.1810±0.0120 | 0.5735±0.0135 |
| | | LoRA | ✘ | ✘ | 0.8946±0.0110 | 0.7021±0.0103 | 0.5517±0.0125 | 0.3669±0.0118 | 0.6625±0.0121 |
| | | QLoRA | ✘ | ✘ | 0.9185±0.0075 | 0.7439±0.0060 | 0.5976±0.0081 | 0.4583±0.0083 | 0.7010±0.0090 |
| | Qwen-72B | SFT | ✘ | ✘ | 0.8313±0.0100 | 0.6345±0.0077 | 0.4886±0.0065 | 0.2772±0.0123 | 0.6033±0.0110 |
| | | LoRA | ✘ | ✘ | 0.9269±0.0075 | 0.7563±0.0059 | 0.6215±0.0083 | 0.3673±0.0136 | 0.7127±0.0094 |
| RAT-SQL | ✘ | ✘ | ✔ | ✘ | 0.8044±0.0107 | 0.6395±0.0082 | 0.5573±0.0124 | 0.4036±0.0101 | 0.6271±0.0119 |
| | BERT-Large | SFT | ✔ | ✘ | 0.8643±0.0119 | 0.7367±0.0145 | 0.6210±0.0093 | 0.4279±0.0116 | 0.6955±0.0124 |
| LGESQL | ✘ | ✘ | ✔ | ✘ | 0.8633±0.0097 | 0.6952±0.0065 | 06154±0.0093 | 0.4106±0.0118 | 0.6768±0.0109 |
| | BERT-Large | SFT | ✔ | ✘ | 0.9150±0.0103 | 0.7647±0.0065 | 0.6673±0.0107 | 0.4888±0.0078 | 0.7421±0.0096 |
| Graphix-T5 | T5-Large | SFT | ✔ | ✘ | 0.8993±0.0075 | 0.7874±0.0068 | 0.5980±0.0102 | 0.4401±0.0083 | 0.7263±0.097 |
| | T5-3B | SFT | ✔ | ✘ | 0.9193±0.0038 | 0.8164±0.0062 | 0.6157±0.0053 | 0.5006±0.0081 | 0.7562±0.0065 |
| RESDSQL | T5-Base | SFT | ✔ | ✘ | 0.9190±0.0047 | 0.8369±0.0051 | 0.6841±0.0070 | 0.5183±0.0065 | 0.7797±0.0073 |
| | T5-Large | SFT | ✔ | ✘ | 0.9355±0.0040 | 0.8543±0.0051 | 0.7241±0.0070 | 0.5361±0.0045 | 0.8008±0.0063 |
| | T5-3B | SFT | ✔ | ✘ | 0.9476±0.0081 | 0.8767±0.0104 | 0.7299±0.0120 | 0.5602±0.0094 | 0.8182±0.0100 |
| DTS-SQL | DeepSeek-7B | SFT | ✘ | ✔ | 0.9274±0.0091 | 0.9013±0.0075 | 0.7414±0.0090 | 0.5663±0.0103 | 0.8269±0.0094 |
| CodeS | CodeLlama-13B | SFT | ✘ | ✔ | 0.9274±0.0084 | 0.8789±0.0052 | 0.7069±0.0079 | 0.5904±0.0038 | 0.8150±0.0070 |
| C³-SQL | GPT-3.5 | ✘ | ✘ | ✔ | 0.9136±0.0068 | 0.8402±0.0094 | 0.7731±0.0064 | 0.6153±0.0080 | 0.8108±0.0095 |
| DIN-SQL | GPT-4 | ✘ | ✘ | ✔ | 0.9234±0.0059 | 0.8744±0.0080 | 0.7644±0.0091 | 0.6265±0.0103 | 0.8279±0.0098 |
| DAIL-SQL | GPT-4 | ✘ | ✘ | ✔ | 0.9153±0.0103 | 0.8924±0.0125 | 0.7701±0.0098 | 0.6024±0.0107 | 0.8308±0.0110 |
| EPI-SQL | GPT-4 | ✘ | ✘ | ✔ | 0.9310±0.0121 | 0.9053±0.0085 | 0.8178±0.0108 | 0.6189±0.0097 | 0.8511±0.0114 |
| SuperSQL | GPT-4 | ✘ | ✘ | ✔ | 0.9435±0.0074 | 0.9126±0.0050 | 0.8333±0.0062 | 0.6867±0.0055 | 0.8682±0.0068 |
| PURPLE | GPT-4 | ✘ | ✘ | ✔ | 0.9404±0.0086 | **0.9206±0.0041** | 0.8268±0.0055 | 0.6715±0.0080 | 0.8670±0.0072 |
| SGU-SQL | GPT-4 | ✘ | ✔ | ✔ | 0.9352±0.0061 | 0.9190±0.0043 | **0.8437±0.0045** | **0.7213±0.0067** | **0.8795±0.0063** |

Table 1: The Execution Accuracy of text-to-SQL models on SPIDER. The best and second-best results in each column are highlighted in **bold** font and underlined. ✔ and ✘ represent that the case is applicable and not applicable, respectively.

**Datasets** We assess the performance of text-to-SQL models using two renowned datasets, Spider (Yu et al., 2019) and BIRD (Li et al., 2023c). Spider, a cross-domain text-to-SQL dataset, comprises 8659 instances in the training split and 1034 instances in the development split, spanning across 200 databases. Each instance comprises a natural language question related to a specific database and its corresponding SQL query. For evaluation purposes, we utilize the Spider-dev development split since the test split has not been released. On the other hand, BIRD (BIg Bench for large-scale Database Grounded text-to-SQL Evaluation) is another pioneering cross-domain dataset that focuses on exploring the impact of extensive database contents on text-to-SQL parsing. BIRD features over 12,751 unique question-SQL pairs, encompassing 95 large databases with a total size of 33.4 GB. It encompasses more than 37 professional domains.

**Baselines** To valid the effectiveness of `SGU-SQL`, we compare it with several state-of-art baselines. Following the taxonomy in Section B, we divide all baselines into three categories: $(i)$ *Fine-tuning*: **T5-base** (Raffel et al., 2020), **T5-large** (Raffel et al., 2020); $(ii)$ *structure-learning*: **RAT-SQL** (Wang et al., 2019), **RASAT** (Qi et al., 2022), **S$^2$SQL** (Hui et al., 2022) ,**RESDSQL** (Li et al., 2023a),**GRAPHIX** (Li et al., 2023b); and $(iii)$ *incontext-learning*: **PaLM-2** (Anil et al., 2023), **CodeX** (Chen et al., 2021), **GPT-4** (OpenAI, 2023), **C3-GPT** (Dong et al., 2023), **DIN-SQL** (Pourreza & Rafiei, 2023), **DAIL-SQL** (Gao et al., 2023), **EPI-SQL** (Liu & Tan, 2024), **SuperSQL** (Li et al., 2024a), **E-SQL** Caferoğlu & Ulusoy (2024), **MAC-SQL** (Wang et al., 2024), **PURPLE** (Ren et al., 2024), **CHESS** Talaei et al. (2024), **CHASE-SQL** Pourreza et al. (2024).

**Evaluation Metrics** We evaluate our models using three key metrics: Exact-Set-Match Accuracy (EM Acc), Execution Accuracy (Exec Acc), and Valid Efficiency Score (VES). EM Acc compares each predicted clause to the validated SQL query, but may produce false results due to value omission. Exec Acc compares execution results of predicted and confirmed SQL queries, offering a more comprehensive assessment by acknowledging multiple valid SQL solutions for a single question. VES measures the efficiency of generated SQLs that produce correct result sets, discounting those that fail to retrieve accurate values. This metric combines execution efficiency and accuracy to provide a holistic performance evaluation.

*Obs.2. In-context learning-based method is better than the methods of the other two categories.* Among the three categories of methods, in-context learning-based methods consistently demonstrate superior performance. This suggests that leveraging in-context learning mechanisms is crucial for enhancing the understanding and generation of SQL queries from natural language inputs. Specifically, the in-context learning-based methods, i.e., DIN-SQL and DAIL-SQL in our comparison set achieve higher accuracy rates and require less computational overhead compared to fine-tuning and structure-learning-based methods. Additionally, the in-context learning-based methods exhibit better generalization across different datasets, indicating their robustness and adaptability.

## 4.2 ABLATION STAUDY: **Q2**

**The effect of prompting strategy** In this part, we conduct comprehensive experiments to investigate the effectiveness of our proposed prompting strategy. Specifically, we compare the structure-based decomposing strategy used in our `SGU-SQL` with other prompting strategies like CoT (Wei et al., 2022) and few-shot prompting. As shown in Table 4 and 8, we can have the following observations.

*Obs.1. Our structure-based decomposing significantly outperforms other simple prompting strategies.* Our method demonstrates superior performance across all tested LLMs. Specifically, compared to CoT, our approach achieves an average improvement of 5.03%, while outperforming few-shot prompting by 4.98% on average.

*Obs.2. Our structure-based decomposing significantly outperforms other advanced prompting strategies. The key distinction of our approach is that it dynamically decomposes queries based on their syntax structure, rather than either using fixed decomposition patterns (like DIN-SQL) or purely relying on LLM's black-box understanding (like ACT-SQL, MAC-SQL).* This syntax-aware decomposition strategy proves more effective for handling complex SQL generation tasks.

*Obs.3. Simple decomposing-based methods are ineffective in the text-to-SQL task.* While decomposing complex tasks into subtasks like CoT, can enhance model performance in many natural language understanding tasks, it proves to be ineffective in the text-to-SQL task. As shown in Table 4 and 8, applying COT on PaLM-2 even leads to a performance decrease of 1.08% compared to the naive few-shot prompting. This is attributed to the complex syntax of SQL, and the intricate correspondence between query terms in user queries and database data units. Conversely, we formally define the meta-operations in SQL and propose a decomposing strategy according to the syntax tree to separate the query into subtasks. This boosts the LLMs' comprehension of linked queries to generate accurate SQLs step by step.

**The generalization ability of prompts**

| Dataset | | Spider | | | BIRD | | |
|---|---|---|---|---|---|---|---|
| Metric | | EX Acc | EM Acc | VES | EX Acc | EM Acc | VES |
| Finetuning-based | Baichuan2-7B | 0.6035 | 0.5793 | 0.6082 | 0.1719 | 0.0547 | 0.2097 |
| | Baichuan2-13B | 0.6776 | 0.6078 | 0.6545 | 0.1766 | 0.0455 | 0.2126 |
| | LlaMA2-7B | 0.6083 | 0.5816 | 0.5795 | 0.1675 | 0.0469 | 0.1670 |
| | LlaMA2-13B | 0.6809 | 0.6400 | 0.6712 | 0.1993 | 0.0743 | 0.1739 |
| | LlaMA2-70B | 0.6869 | 0.6555 | 0.6779 | 0.2414 | 0.0778 | 0.1987 |
| | CodeLlama-7B | 0.7018 | 0.6431 | 0.7357 | 0.2370 | 0.1283 | 0.2504 |
| | CodeLlama-13B | 0.7462 | 0.7056 | 0.7391 | 0.2944 | 0.2551 | 0.3004 |
| | CodeLlama-70B | 0.7710 | 0.7139 | 0.7463 | 0.3287 | 0.2557 | 0.3428 |
| | Qwen-7B | 0.6519 | 0.6106 | 0.6625 | 0.1709 | 0.0439 | 0.1915 |
| | Qwen-14B | 0.6625 | 0.6238 | 0.6757 | 0.2286 | 0.0645 | 0.2396 |
| | Qwen-72B | 0.7127 | 0.6812 | 0.7082 | 0.2392 | 0.0894 | 0.2488 |
| Structure Learning | RAT-SQL | 0.6955 | 0.6597 | 0.6734 | 0.2639 | 0.2431 | 0.2431 |
| | BRIDGE | 0.6928 | 0.7053 | 0.6893 | 0.2459 | 0.2068 | 0.2574 |
| | LGESQL | 0.7421 | 0.7251 | 0.7067 | 0.2837 | 0.2493 | 0.2889 |
| | S$^2$SQL | 0.7643 | 0.7385 | 0.7539 | 0.2960 | 0.2649 | 0.3143 |
| | RESDSQL | 0.8182 | 0.7580 | 0.8226 | 0.3312 | 0.3174 | 0.3286 |
| | Graphix-T5 | 0.7562 | 0.7463 | 0.7643 | 0.2984 | 0.2538 | 0.3062 |
| | METASQL | 0.7695 | 0.7288 | 0.7498 | 0.3180 | 0.3011 | 0.3225 |
| In-Context Learning | GPT-3.5 | 0.7394 | 0.5327 | 0.7457 | 0.3562 | 0.3041 | 0.3415 |
| | GPT-4 | 0.7665 | 0.5892 | 0.7390 | 0.4633 | 0.4255 | 0.4794 |
| | PaLM-2 | 0.6985 | 0.4438 | 0.7148 | 0.2735 | 0.2543 | 0.3061 |
| | CodeX | 0.7167 | 0.4905 | 0.7011 | 0.3438 | 0.3019 | 0.3496 |
| | C$^3$-GPT | 0.8108 | 0.7036 | 0.8009 | 0.5020 | 0.4143 | 0.5077 |
| | DIN-SQL | 0.8279 | 0.7187 | 0.8173 | 0.5072 | 0.4398 | 0.5879 |
| | DAIL-SQL | 0.8308 | 0.7443 | 0.8317 | 0.5434 | 0.4581 | 0.5576 |
| | DTS-SQL | 0.8269 | 0.7260 | 0.8163 | 0.5581 | 0.4825 | 0.6038 |
| | CodeS | 0.8150 | 0.7069 | 0.8092 | 0.5714 | 0.4893 | 0.6120 |
| | SuperSQL | 0.8682 | 0.7589 | 0.8410 | 0.5860 | 0.4745 | 0.6067 |
| | MAC-SQL | 0.8635 | 0.7545 | 0.8541 | 0.5759 | 0.4906 | 0.5872 |
| | SGU−SQL | **0.8795** | **0.7826** | **0.8652** | **0.6180** | **0.5144** | **0.6393** |

Table 2: The Execution Accuracy and Exact Match Accuracy of text-to-SQL models on SPIDER and BIRD. The best and second-best results in each column are highlighted in **bold** font and underlined. NaN denotes that the result is not available.

To further verify the generalization ability of our proposed prompting strategy, in this part, we conduct comprehensive experiments to investigate whether SGU−SQL could enhance other LLMs by substituting their original framework with the decomposing-based prompts. Specifically, we replace GPT-4 used in SGU−SQL with other representative generative LLMs, including PaLM-2 (Anil et al., 2023), CodeX (Chen et al., 2021), ChatGPT and GPT-4 (OpenAI, 2023) as alternatives. Specifically, we used the model 'chat-bison-001' provided by GoogleAI as the implementation of PaLM-2, and 'ChatGPT-turbo' and 'gpt-4' as the implementations of ChatGPT and GPT-4, respectively. The text-to-SQL task is conducted under the few-shot setting with the query from the development set of Spider as input. As shown in Figure 3, we have the following observations.

*Obs.1. The performances of the original LLMs improved significantly by integrating the prompt learned from our* SGU−SQL. Specifically, PaLM-2 improved by 4%, CodeX by 3%, ChatGPT by 5%, and GPT-4 by almost 11%. The substantial performance gains indicate the robustness and generalization ability of our proposed prompting strategy. Furthermore, the consistent improvements across different LLMs highlight the versatility and applicability of our approach in enhancing the capabilities of existing language models.

*Obs.2. LLMs with stronger reasoning abilities exhibit greater improvement.* We observe that LLMs with stronger reasoning abilities benefit more from integrating the prompts learned from SGU−SQL.

Specifically, GPT-4, which is known for its advanced reasoning capabilities, shows a more substantial performance improvement compared to PaLM-2, CodeX, and ChatGPT. This suggests that our prompting strategy is particularly effective in enhancing the performance of LLMs that require more complex reasoning tasks.

### 4.3 Model Analysis

**Difficulty analysis Q3:** In this part, we first analyze the performance of our proposed method on queries with different levels of difficulty. Our analysis focused on evaluating the performance of our proposed method across queries of varying difficulty levels. Table 1 provides a comparative assessment of our method against state-of-the-art (SOTA) prompting methods on the Spider development set. Our findings reveal that our method consistently outperforms competing methods across all difficulty levels. Notably, we observe the most substantial improvements in the extra hard and hard classes, where other prompting models struggle. Additionally, our method also shows a slight improvement in the easy class, which suggests that our method is robust and effective across queries of different difficulty levels, highlighting its potential for practical applications in natural language understanding and query generation tasks.

**Error analysis Q4:** We checked the errors in the generated SQL answers and classified them into six categories, as shown in Figure 4 following the classification by (Pourreza & Rafiei, 2023). We discuss the failure cases of our model in comparison with baseline models and then discuss the reasons for the typical failure of LLMs in Text-to-SQL tasks. Compared to the baseline model, we achieved a reduction of approximately 33.5% in errors and made progress in the schema-linking and join statement components where traditional models often falter. In this section, we will first discuss the failure cases of our model in comparison with baseline models, and then discuss the reasons for the typical failure for LLMs in text-sql tasks. We checked the errors in the generated SQL answers and classified them into six categories, as shown in Figure 4 followed by the major classification by (Pourreza & Rafiei, 2023). Compared to the baseline model, we achieved a reduction of approximately 33.5% in errors and made progress in the schema-linking and join statement components where traditional models often falter. Errors in the schema-linking segment decreased by around 38%, primarily attributed to the utilization of Precise Query Matching, wherein graph neural networks were employed to learn and match the database schema. This underscores the efficacy of Structure Linking. In the sections prone to errors, such as Group-by and Join, our errors decreased by 35%, indicating that our syntax tree decomposing enables the model to more accurately utilize corresponding SQL Meta-operations to achieve the intention queries, thus further enhancing the accuracy in identifying the targeted tables or columns for manipulation.

## 5 Conclusion

Recent advancements in large language models (LLMs) have shown promise in improving the accuracy of text-to-SQL generation. However, existing models typically input queries and database schemas into LLMs to perform semantic-structure matching and generate structured SQL, while often overlook the structural information inherent in user queries and databases, which could significantly enhance the generation of accurate SQL queries. This oversight can result in the production of inaccurate or inexecutable SQL queries. To fully exploit the structure, we propose the structure-to-SQL framework (SGU-SQL), which leverages the inherent structure information to improve the SQL generation of LLMs. Specifically, SGU-SQL links user queries and databases in a structure-enhanced manner. It then decomposes complicated linked structures with syntax trees to guide the LLM to generate the SQL step by step. Extensive experiments on two benchmarks demonstrate that SGU-SQL consistently outperforms state-of-the-art SQL generation baselines. These results highlight the importance of explicitly incorporating structural information for effective text-to-SQL generation.

### Ethics and Reproducibility Statements

Our research adheres to high ethical standards and all experiments were conducted using publicly available datasets, which are clearly cited in the paper. The code for our models and experiments will be made available in a public GitHub repository upon acceptance of the paper. This repository will provide detailed instructions, including environment setup, and scripts to reproduce our experiments.

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

# A PRELIMINARIES

## A.1 STRUCTURE LEARNING FOR TEXT-TO-SQL

**Definition 1. Structure Learning for Text-to-SQL**: Given a natural language query $\mathcal{D}$ and a database schema $\mathcal{Q}$, the task of graph learning for Text-to-SQL aims to generate a graph-based representation $\mathcal{G}$ that captures the structural and semantic relationships between the query and the schema, and to learn a mapping function $f : \mathcal{G}_q \rightarrow \mathcal{G}_d$, where $\mathcal{G}_q$ is the structural user queries, and $\mathcal{G}_d$ is the corresponding database contents linked to the query $\mathcal{G}_q$.

Let $\mathcal{G} = (\mathcal{V}, \mathcal{E})$ denote the graph representation, where $\mathcal{V}$ is the set of nodes and $\mathcal{E}$ is the set of edges. The nodes $v \in \mathcal{V}$ represent the entities and components in the query and schema, such as tables, columns, and query tokens. The edges $e \in \mathcal{E}$ represent the relationships and dependencies between the nodes. The graph learning task involves two main components, including graph construction and graph representation learning.

### A.1.1 GRAPH CONSTRUCTION

The first step is to construct the graph $\mathcal{G}$ from the query $\mathcal{Q}$ and schema $\mathcal{D}$. This involves extracting relevant entities and relationships from the input and organizing them into a graph structure. The graph construction process can be formally defined as:

$$\mathcal{G} = \text{Construct}(Q, D), \tag{19}$$

where $\text{Construct}(\cdot)$ is a method that maps the query and schema to the graph representation.

### A.1.2 GRAPH REPRESENTATION LEARNING

Once the graph is constructed, the next step is to learn meaningful representations of the nodes and edges in the graph. This is typically achieved using Graph Neural Networks (GNNs), which propagate information across the graph structure to capture the structural and semantic relationships. The representation learning process can be formally defined as:

$$\mathbf{h}_v^{(l+1)} = \text{GNN}(\mathbf{h}_v^{(l)}, \{\mathbf{h}_u^{(l)} : u \in \mathcal{N}(v)\}), \tag{20}$$

where $\mathbf{h}_v^{(l)}$ is the representation of node $v$ at layer $l$, $\mathcal{N}(v)$ is the set of neighboring nodes of $v$, and $\text{GNN}(\cdot)$ is the graph neural network function that updates the node representations based on their neighbors. The learned graph representations are then used to generate the corresponding SQL query $\mathcal{S}$ by applying a decoding function $f$ to the graph:

$$S = f(\mathcal{G}). \tag{21}$$

The objective of graph learning for Text-to-SQL is to optimize the parameters of the graph construction and representation learning components, as well as the decoding function, to generate accurate and executable SQL queries from natural language queries and database schemas.

## A.2 TEXT-TO-SQL GENERATION WITH LLMS

We now formally define the problem of text-to-SQL generation. Let $\mathcal{D}$ be a database schema consisting of a set of tables $\mathcal{T} = \{T_1, T_2, \ldots, T_n\}$, where each table $T_i$ has a set of columns $\mathcal{C}_i = \{C_{i1}, C_{i2}, \ldots, C_{im}\}$. The database schema $\mathcal{D}$ can be represented as a tuple $(\mathcal{T}, \mathcal{C})$, where $\mathcal{C} = \bigcup_{i=1}^n \mathcal{C}_i$ is the set of all columns across all tables.

Given a natural language query $Q$ and a database schema $\mathcal{D}$, the task of Text-to-SQL generation aims to translate $Q$ into a corresponding SQL query $S$ that accurately retrieves the desired information from the database. Let $\mathcal{M}$ be the LLM that maps the natural language query $Q$ and the database schema $\mathcal{D}$ to the target SQL query $S$, the main objective can be formulated as follows:

$$\mathcal{M} : (Q, \mathcal{D}, \theta) \rightarrow S. \tag{22}$$

The objective of LLM-based text-to-SQL generation is to learn the optimal parameters or prompts $\theta^*$ that minimize the difference between the generated SQL query $\mathcal{M}(Q, \mathcal{D}, \theta)$ and the ground truth SQL query $S$:

$$\theta^* = \arg\min_\theta \mathcal{L}(\mathcal{M}(Q, \mathcal{D}, \theta), S), \tag{23}$$

where $\mathcal{L}$ is a loss function that measures the discrepancy between generated and ground truth SQLs.

## B RELATED WORK

Text-to-SQL has witnessed significant evolution over the past few years. Early researchers focused on well-designed rules, which were later superseded by deep learning-based techniques. More recently, the integration of pre-trained language models (PLMs) and large language models (LLMs) has further advanced state-of-the-art text-to-SQL generation. This section traces the developmental trajectory of Text-to-SQL methods, highlighting the key milestones and innovations that have shaped the field.

### B.1 TRADITIONAL TEXT-TO-SQL METHODS

Text-to-SQL has witnessed significant advancements in recent years. Early research heavily relied on well-designed rules and templates (Li & Jagadish, 2014; Mahmud et al., 2015; Yu et al., 2021), which were suitable for simple database scenarios. However, the increasing complexity of database structure and the high labor costs associated with rule-based methods have made such approaches impractical. The advent of deep neural networks, such as sequence-to-sequence models and encoder-decoder structures like LSTMs (Hochreiter & Schmidhuber, 1997) and Transformers (Vaswani et al., 2017), has revolutionized the field of text-to-SQL (Guo et al., 2019; Choi et al., 2021). They automatically learn a mapping from user queries to corresponding SQL queries. Typically, RYANSQL (Choi et al., 2021) introduced intermediate representations and sketch-based slot filling to handle complex questions and improve cross-domain generalization. More recently, pre-trained language models

---

**Algorithm 1** Syntax-based Subtask Decomposition for Text-to-SQL Generation

---

**Require:** Context-free syntax tree $\mathcal{T}$, non-terminal nodes $N$, production rules $R$, mapping function $f : N \to S$

**Ensure:** SQL query $S$

1: **function** GENERATESQL($\mathcal{T}, N, R, f$)
2:     $S \leftarrow \emptyset$                                       ▷ Initialize the SQL query
3:     $n_0 \leftarrow$ root node of $\mathcal{T}$
4:     TRAVERSETREE($n_0, S$)
5:     **return** $S$
6: **end function**
7: **function** TRAVERSETREE($n, S$)
8:     **if** $n \in N$ **then**                          ▷ Check if $n$ is a non-terminal node
9:        $s_n \leftarrow f(n)$                     ▷ Generate SQL component for node $n$
10:       $S \leftarrow S \cup s_n$                    ▷ Add SQL component to the query
11:       $r \leftarrow$ production rule that expands $n$
12:       **for** each child node $c$ of $n$ **do**
13:          TRAVERSETREE($c, S$)            ▷ Recursive traversal of child nodes
14:       **end for**
15:       COMBINESQL($n, r, S$)     ▷ Combine SQL components based on production rule
16:     **end if**
17: **end function**
18: **function** COMBINESQL($n, r, S$)
19:     $s_n \leftarrow$ SQL component corresponding to node $n$
20:     $s_{c_1}, s_{c_2}, \ldots, s_{c_k} \leftarrow$ SQL components of child nodes of $n$
21:     $s_{combined} \leftarrow$ Combine($s_{c_1}, s_{c_2}, \ldots, s_{c_k}$) based on production rule $r$
22:     $S \leftarrow S \setminus s_n \cup s_{combined}$                       ▷ Update the SQL query
23: **end function**

---

(PLMs) with strong semantic parsing capabilities have become the new paradigm of text-to-SQL systems. The initial adoption of PLMs in Text-to-SQL primarily focused on fine-tuning off-the-shelf models, such as BERT (Devlin et al., 2019) and RoBERTa, on standard text-to-SQL datasets (Yu et al., 2018; Zhong et al., 2017). Incremental research on PLM-based optimization, such as table content encoding (Guo et al., 2019; Yin et al., 2020; Dou et al., 2022). and schema information incorporation (Li et al., 2023a), has further advanced this field.

### B.2 LLM-BASED TEXT-TO-SQL MODELS

Large language models (LLMs), such as GPT series (Radford et al., 2018; Brown et al., 2020; Achiam et al., 2023), have gained significant attention in recent years due to their capability to generate coherent and fluent text. Researchers have started exploring the potential of LLMs for text-to-SQL by leveraging their extensive knowledge reserves and superior generation capabilities (Rajkumar et al., 2022; Gao et al., 2024). These approaches often involve fine-tuning the open-source LLMs on text-to-SQL datasets (Anil et al., 2023; Hong et al., 2024) or prompt engineering to guide the closed-source LLMs in SQL generation (Chang & Fosler-Lussier, 2023; Pourreza & Rafiei, 2023; Gao et al., 2024).

### B.2.1 FINE-TUNING LLMS FOR TEXT-TO-SQL

Recently, the emergence of large language models (LLMs) has markedly altered the landscape for text-to-SQL tasks. LLMs, with their capacity for understanding and generating human-like text, present a robust solution for text-to-SQL applications (Liu et al., 2023). The development of LLMs typically encompasses pre-training followed by fine-tuning. Research has concentrated on fine-tuning with domain-specific data and optimization techniques to enhance base models for coding tasks, including text-to-SQL. This process enables models to master programming language syntax and database schema intricacies (Raffel et al., 2020; Roziere et al., 2023). Through training on tailored datasets of annotated SQL queries, LLMs acquire the syntax and structure necessary for generating

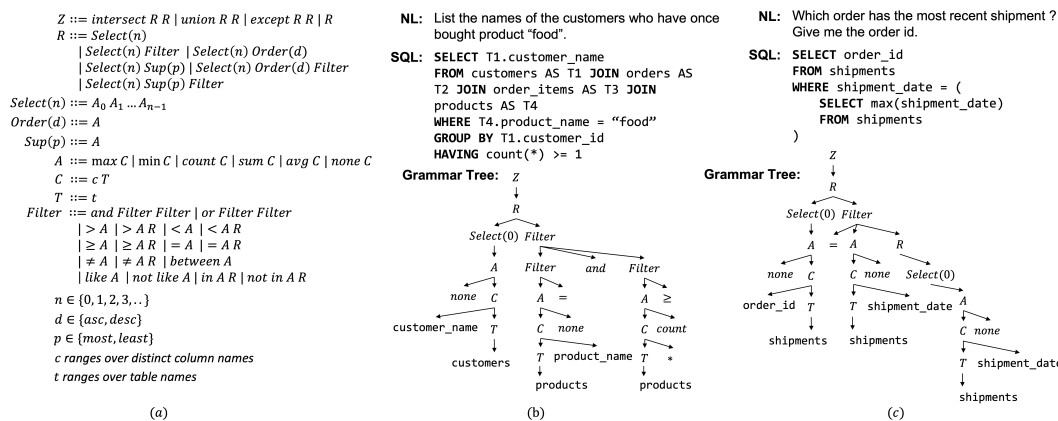

Figure 2: The example of syntax tree. Subfigure $(a)$ denotes the context-free grammar rule of SQL. While subfigures $(b)$ and $(c)$ demonstrate two examples of the constructed syntax tree used in SGU-SQL.

compliant SQL code (Trummer, 2022; Sun et al., 2023). Furthermore, PICARD (Scholak et al., 2021) introduced a decoding mechanism for LLMs that ensures the generation of valid sequences by discarding inadmissible tokens at each step, employing incremental parsing to guarantee the validity of SQL queries produced by autoregressive language models. More recently, data-augmented fine-tuning techniques have emerged as a promising approach to improve text-to-SQL generation models. By focusing on enhancing the quality and diversity of the training data during supervised fine-tuning, these methods enable models to better capture the complexities of translating natural language queries into SQL statements. For example, Symbol-LLM (Xu et al., 2024) proposes a two-stage approach, consisting of an injection stage and an infusion stage, for data-augmented instruction tuning. This method effectively incorporates additional data to improve the LLM's ability to follow instructions. Similarly, CodeS (Li et al., 2024b) leverages ChatGPT to generate bi-directional training data, augmenting the model's training dataset and enhancing its code generation capabilities. Additionally, StructLM (Zhuang et al., 2024) introduces a training paradigm that involves multiple structured knowledge tasks, aiming to improve the model's overall performance across a wide range of applications. These approaches demonstrate the potential of data augmentation and multi-task learning in boosting the performance of LLMs.

### B.2.2 IN-CONTEXT LEARNING FOR TEXT-TO-SQL

In-context learning enhances LLM performance by providing detailed task instruction, background knowledge, and contextual examples during inference, thereby improving performance for specific tasks. This approach has seen innovative applications in text-to-SQL, with strategies aimed at optimizing prompt contents and formats based on user queries and database structures. Typically, C3-SQL (Dong et al., 2023) designed a zero-shot prompting framework for ChatGPT with clear prompting for effective input format and tailored hints for calibration and consistency checking during the query generation. KATE (Liu et al., 2021) first investigated the impact of few-shot examples on GPT-3's performance. (Nan et al., 2023) further conducted a systematic investigation into different demonstration selection methods and optimal instruction formats for prompting LLMs in the text-to-SQL task, whereas DESEM (Guo et al., 2023) developed a domain-specific vocabulary masking technique, called similarity assessment, highlighting the relevance of SQL-specific terms. DIN-SQL (Pourreza & Rafiei, 2023) introduced a decomposed framework, categorizing user queries by complexity and breaking down the generation task into sub-problems and feeding the solutions of those sub-problems into LLMs to improve the generation performance of complex SQL queries. DAIL-SQL (Gao et al., 2024) further enhanced the performance by incorporating suitable formatting of the database schema and selecting examples based on skeleton similarities. Some recent work improves the in-context learning framework by incorporating execution feedback through second-round prompting for regeneration. For example, MRC-EXEC (Shi et al., 2022) introduced a natural language to code translation framework with execution, which executes each sampled SQL query and selects the example with the minimal execution result–based Bayes risk (Müller & Sennrich, 2021). LEVER (Ni et al., 2023) proposed an approach to verify NL2Code with execution, utilizing a

generation and execution module to collect sampled SQL set and their execution results, respectively, then using a learned verifier to output the probability of the correctness. Similarly, the SELF-DEBUGGING (Chen et al., 2024) framework is presented to teach LLMs to debug their predicted SQL via few-shot demonstrations. The model can refine its mistakes by investigating the execution results and explaining the generated SQL in natural language without human intervention.

## B.3   STRUCTURE LEARNING FOR TEXT-TO-SQL

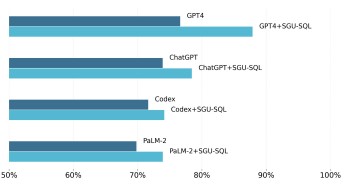

Figure 3: Ablation Study of SGU-SQL - EX Acc on SPIDER.

Structure learning-based models, particularly those utilizing Graph Neural Networks (GNNs), have emerged as a powerful approach to modeling the complex relationships between user queries and database schemas in text-to-SQL generation. By organizing information into graph structures and leveraging GNNs to learn rich structural representations, these methods enhance the semantic understanding and generalization ability of text-to-SQL models. Specifically, RATSQL (Wang et al., 2019) employs a graph-based structure to delineate relationships within database schemas and queries, treating the schema as a graph of tables and columns connected by relational edges. LGESQL (Cao et al., 2021) introduced an edge-centric graph model derived from conventional node-centric graphs, to capture diverse structural topologies. $S^2$SQL (Hui et al., 2022), integrates syntactic dependency information into a question-schema interaction graph, focusing on primary relationships to mitigate overfitting while emphasizing essential graph structures. Graphix-T5 (Li et al., 2023b) explored the integration of GNN layers into the large language model T5 (Raffel et al., 2020), aiming to leverage both semantic and structural information from PLMs and GNNs, respectively. RESDSQL (Li et al., 2023a) designed a ranking-enhanced encoder to rank and filter the schema items for skeleton-aware schema linking and the skeleton parsing.

## C   ABLATION STUDY

In this section, we have conducted detailed experiments to validate the effectiveness of each component in SGU-SQL.

## C.1   THE EFFECT OF STRUCTURE LEARNING

As shown in Table 3, we have the following observations: $(i)$ Removing the query graph representation leads to significant performance drops (-3.45% on Spider-dev, -2.87% on BIRD-dev), demonstrating that our proposed query graph is crucial for understanding the intent behind the query. $(ii)$ The ablation of the database graph results in performance decreases of -2.14% on Spider-dev and -3.54% on BIRD-dev. The larger performance drop on BIRD (-3.54%) vs Spider (-2.14%) indicates that graph-based database representation is particularly important for complex, realistic databases. $(iii)$ When removing structure-aware linking, we observe substantial performance degradation (-5.33% on Spider-dev, -6.49% on BIRD-dev), representing the second-largest impact among all components. The more significant drop on BIRD emphasizes that our linking mechanism is particularly crucial for complex queries and databases, effectively bridging the semantic gap between natural language and database components while maintaining structural integrity.

| Variant | Full Model | w/o query graph | w/o database graph | w/o structure linking | w/o decomposition |
|---|---|---|---|---|---|
| SPIDER-dev | 87.95 | 84.50 (-3.45) | 85.81 (-2.14) | 82.62 (-5.33) | 82.35 (-5.60) |
| BIRD-dev | 61.80 | 58.93 (-2.87) | 58.26 (-3.54) | 55.31 (-6.49) | 53.78 (-8.02) |

Table 3: Ablation study on different components of SGU-SQL.

## C.2 THE EFFECT OF SYNTAX-BASED DECOMPOSITION

To verify the effectiveness of our syntax-based decomposition strategy, we conducted additional experiments to compare our SGU-SQL with other advanced decomposition-based methods, including DIN-SQL, ACT-SQL and MAC-SQL.

| Text-to-SQL | DIN-SQL | ACT-SQL | MAC-SQL | SGU-SQL |
|---|---|---|---|---|
| SIPDER-dev | 82.79 | 82.90 | 86.35 | 87.95 |

Table 4: Performance comparison between SGU-SQL and advanced decomposition methods.

As shown in Tables 3 and 4, $(i)$ the ablation of our decomposition strategy leads to the most significant performance decrease (-5.60% on Spider-dev, -8.02% on BIRD-dev). These results validate our approach of breaking down complex queries into manageable components while preserving structural relationships, especially beneficial for real-world applications involving complex dabase structure and intricate query patterns. $(ii)$ Our SGU-SQL achieves 87.95% execution accuracy on SPIDER-dev, outperforming all these methods. The key distinction of our approach is that it dynamically decomposes queries based on their syntax structure, rather than either using fixed decomposition patterns (like DIN-SQL) or purely relying on LLM's black-box understanding (like ACT-SQL, MAC-SQL). This syntax-aware decomposition strategy proves more effective for handling complex SQL generation tasks.

## C.3 THE EFFECT OF BACKBONE LLMs

For a thorough evaluation of SGU-SQL's performance, we conduct additional experiments on BIRD dev with different LLMs as backbones. Specifically, we compared SGU-SQL against two categories of methods: $(i)$ Open-source models with available paper and codes: MAC-SQL, Super-SQL, E-SQL and CHESS; and $(ii)$ Undisclosed methods that have demonstrated strong performance: PURPLE, Distillery and CHASE-SQL.

| Backbone | MAC-SQL | PURPLE | E-SQL | CHESS | Distillery | CHASE-SQL | SGU-SQL |
|---|---|---|---|---|---|---|---|
| GPT-4 | 59.59 | 60.71 | 58.95 | 61.37 | - | - | 61.80 |
| GPT-4o | 65.05 | 68.12 | 65.58 | 68.31 | 67.21 | - | 69.28 |
| Gemini-1.5 Pro | - | - | - | - | - | 73.14 | - |

Table 5: Performance comparison on BIRD dev with different LLMs as backbones.

As shown in Table 5, our SGU-SQL achieves competitive performance across different LLM backbones. Specifically, we have the following observations: Using GPT-4 as the backbone, SGU-SQL achieves the best performance compared to other models using the same backbone. With GPT-4o, SGU-SQL achieves 69.28% in terms of execution accuracy, outperforming several strong baselines: PURPLE (68.12%), CHESS (68.31%), E-SQL (65.58%) and Distillery (67.21%). The only model showing higher performance is CHASE-SQL (released in October 2024), which uses Gemini 1.5 Pro as its backbone. Notably, CHASE-SQL incorporates a query fixer module that leverages database execution feedback to guide LLMs to iteratively refine generated queries. In contrast, our model generates SQL queries in a single pass without utilizing any execution feedback.

# D MODEL ANALYSIS

## D.1 PERFORMANCE ON MORE CHALLENGING DATASET

To further verify the effectiveness of our model, we conduct additional experiments on more challenging datasets, like Spider 2.0-Snow and Spider 2.0-Lite Lei et al. (2024). As shown in Table 6, while the performances are relatively low across all models, SGU-SQL consistently demonstrates better capability in handling complex SQL generation tasks in both single and multi-database scenarios.

## D.2 EFFICIENCY ANALYSIS

To assess our approach thoroughly, we conducted the efficiency analysis on the BIRD dataset, a large-scale benchmark in text-to-SQL research with 12,751 unique question-SQL pairs across 95 databases (33.4 GB total). Given that the queries in this dataset are categorized into 3 difficulty levels: simple, moderate, and challenging, we specifically tested our model on the challenging set of the BIRD dataset and compared its performance with DIN-SQL and MAC-SQL.

As shown in Table 7, our model demonstrates superior performance while maintaining competitive computational efficiency. Specifically, our model requires less time for both training and inference. This superior efficiency can be attributed to our graph-based architecture. While baseline methods avoid the overhead of graph construction, they heavily rely on prompt-based modules that require multiple calls to LLMs like GPT-4. These API calls introduce substantial latency that accumulates during both the training and inference phases. In contrast, our graph-based approach, despite its initial graph construction overhead, achieves faster end-to-end processing by minimizing dependence on time-consuming API calls.

## D.3 DIFFICULTY ANALYSIS Q3

In this part, we first analyze the performance of our proposed method on queries with different levels of difficulty. Our analysis focused on evaluating the performance of our proposed method across queries of varying difficulty levels. Table 1 provides a comparative assessment of our method against state-of-the-art (SOTA) prompting methods on the Spider development set. Our findings reveal that our method consistently outperforms competing methods across all difficulty levels. Notably, we observe the most substantial improvements in the extra hard and hard classes, where other prompting models struggle. Additionally, our method also shows a slight improvement in the easy class, which suggests that our method is robust and effective across queries of different difficulty levels, highlighting its potential for practical applications in natural language understanding and query generation tasks.

## D.4 ERROR ANALYSIS Q4

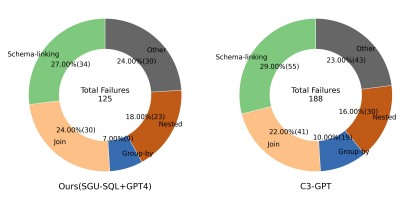

Figure 4: Error Analysis of GPT-4 + SGU-SQL and C3-GPT on the Dev Set: A Comparison of 125 and 188 Failures.

We checked the errors in the generated SQL answers and classified them into six categories, as shown in Figure 4 following the classification by (Pourreza & Rafiei, 2023). We discuss the failure cases of our model in comparison with baseline models and then discuss the reasons for the typical failure of LLMs in Text-to-SQL tasks. Compared to the baseline model, we achieved a reduction of approximately 33.5% in errors and made progress in the schema-linking and join statement components where traditional models often falter. In this section, we will first discuss the failure cases of our model in comparison with baseline models, and then discuss the reasons for the typical failure for LLMs in text-sql tasks. We checked the errors in the generated SQL answers and classified them into six categories, as shown in Figure 4 followed by the major classification by (Pourreza & Rafiei, 2023). Compared to the baseline model, we achieved a reduction of approximately 33.5% in errors and made progress in the schema-linking and join statement components where traditional models often falter. Errors in the schema-linking segment decreased by around 38%, primarily attributed to the utilization of Precise Query Matching, wherein graph neural networks were employed to learn

| Datasets | DAIL-SQL+GPT-4o | CHESS+GPT-4o | SGU-SQL+GPT-4o |
|---|---|---|---|
| Spider 2.0-Snow | 2.20 | 1.28 | 4.39 |
| Spider 2.0-Lite | 5.68 | 3.84 | 6.40 |

Table 6: Execution accuracy for baseline methods on Spider 2.0-Snow and Spider 2.0-Lite.

and match the database schema. This underscores the efficacy of Structure Linking. In the sections prone to errors, such as Group-by and Join, our errors decreased by 35%, indicating that our syntax tree decomposing enables the model to more accurately utilize corresponding SQL Meta-operations to achieve the intention queries, thus further enhancing the accuracy in identifying the targeted tables or columns for manipulation.

To further analyze the reasons for errors in the baseline model, we conducted a comprehensive case study by comparing the results of the baseline model with those of our model, as shown in Figure 5.

**Subtask Decomposing** LLMs often do not adequately break down the task into its essential steps for reasoning. For example, in Case 1, the primary subtask of linking flight data to specific cities was ignored. The question did not adequately break down the task into its essential components without further guidance from LLMs. In Case 3, the query did not decompose the task into two separate subtasks to identify semesters with Masters and Bachelors enrollments independently which also leads to wrong returned answers.

**Intention Understanding** LLMs sometimes misunderstand the core intention of the question. In Case 2, LLMs fail to identify the intention that the question is trying to find all countries where English is spoken, regardless of its official status which leads to errors. It concentrated on the official language status, which did not align with the broader objective of considering English-speaking countries in general. In Case 1, the query was centered around airport codes (SourceAirport), misinterpreting the intention to identify the busiest city, not just the airport. In Case 3, LLM misinterprets the intention of finding how many likes Kyle has received. It erroneously assumes the task is to count how many likes Kyle has given, not received.

**Data Schema Linking** Since LLMs get data schema information with plain text as inputs, it might be challenging to reason the right linking strategy to solve the problem correctly. It needs to understand the referenced tables and columns in the question which are often being mentioned in an inexplicitly way, then matching with the database schema. In contrast, our tailored GNN model can handle this situation well. In Case 1, the initial query failed to incorporate the airport's table, which was essential for linking airport codes to their respective cities. In Case 3, the query did not effectively link degree program types (Masters, Bachelors) to semesters in databases in a way that would allow for the inclusive identification of valid semesters. There was also a misalignment in linking: `student_id` from the `Likes` table was incorrectly associated with the `id` in `Highschooler` table. It should link `liked_id` from `Likes` to `id` in `Highschooler` to align with the task's objective.

# E  FUTURE WORK

Discussing potential extensions is crucial for the research community. Following your suggestion, we have identified several promising future research directions from the following there perspectives.

## E.1  TECHNICAL EXTENSIONS

### E.1.1  STRUCTURE-AWARE FEW-SHOT EXAMPLE SELECTION

While our framework emphasizes the significance of the decomposition strategy, we recognize that the performance of LLM-based text-to-SQL can be further enhanced through tailored few-shot example selection. Current approaches to few-shot example selection primarily rely on keyword matching

| Text-to-SQL | Backbone LLM | Training Time | Inference Time | Performance |
|---|---|---|---|---|
| DIN-SQL | + GPT-4 | 4.69 h | 0.39 h | 36.7% |
| MAC-SQL | + GPT-4 | 4.98 h | 0.36 h | 39.3% |
| SGU-SQL | + GPT-4 | 3.47 h | 0.22 h | 42.1% |

Table 7: Efficiency analysis on the 'Challenging' set of BIRD.

| NL   Query: | *What is the code of airport that has the highest number of flights?* |
|---|---|
| Prompts from Baseline Model | ```/* Given the following database schema:  */
CREATE TABLE 'flights' (
   Airline INTEGER,
   FlightNo INTEGER,
   SourceAirport TEXT,
   DestAirport TEXT,
   PRIMARY KEY(Airline, FlightNo),
   FOREIGN KEY (SourceAirport) REFERENCES airports(AirportCode),
   FOREIGN KEY (DestAirport) REFERENCES airports(AirportCode));
/* Answer the following:  What is the code of airport*/;
that has the highest number of flights?  */;
Let's think step by step.``` |
| Results from Baseline Model's Prompt | ```SELECT SourceAirport, COUNT(*) AS NumberOfFlights
FROM flights GROUP BY SourceAirport
ORDER BY NumberOfFlights DESC LIMIT 1;``` |
| **Gold SQL** | ```SELECT T1.AirportCode
FROM AIRPORTS AS T1
JOIN FLIGHTS AS T2 ON T1.AirportCode = T2.DestAirport
  OR T1.AirportCode = T2.SourceAirport
GROUP BY T1.AirportCode
ORDER BY count(*) DESC LIMIT 1;``` |

(a) Case 1: Airports Database Question

| NL Query: | *How many people live in countries that do not speak English?* |
|---|---|
| Prompts from Baseline Model | ```/* Given the following database schema:  */
CREATE TABLE countrylanguage (
   CountryCode char(3) NOT NULL DEFAULT '',
   Language char(30) NOT NULL DEFAULT '',
   IsOfficial text NOT NULL DEFAULT 'F',
   Percentage float(4,1) NOT NULL DEFAULT '0.0',
   CONSTRAINT countryLanguage_ibfk_1)
   FOREIGN KEY (CountryCode) REFERENCES country (Code));
/* Answer the following:  What is the code of airport
that has the highest number of flights?  */
Let's think step by step.``` |
| Results from Baseline Model's Prompt | ```SELECT SUM(c.Population) FROM country c
WHERE c.Code NOT IN (
   SELECT cl.CountryCode FROM countrylanguage cl
   WHERE cl.Language = 'English' AND
   cl.IsOfficial = 'T' );``` |
| **Gold SQL** | ```SELECT sum(Population) FROM country
WHERE Name NOT IN (
   SELECT T1.Name FROM country AS T1
   JOIN countrylanguage AS T2
   ON T1.Code = T2.CountryCode
   WHERE T2.Language = "English" );``` |

(b) Case 2: Languages Database Question

| NL Query: | *Return the number of likes that the high schooler named Kyle has.* |
|---|---|
| Prompts from Baseline Model | ```/* Given the following database schema:  */
CREATE TABLE Likes(
student_id int,
liked_id int,
primary key (student_id, liked_id),
   foreign key (liked_id) references Highschooler(ID))
   FOREIGN KEY (CountryCode) REFERENCES country (Code))
/* Answer the following:  Return the number of likes that
the high schooler named Kyle has.  */
Let's think step by step.``` |
| Results from Baseline Model's Prompt | ```SELECT count(*)
FROM Likes JOIN Highschooler
  ON Likes.liked_id = Highschooler.ID
  WHERE Highschooler.name = 'Kyle');``` |
| **Gold SQL** | ```SELECT count(*) FROM Likes
   ROM country AS T1 JOIN Highschooler AS T2
   N T1.student_id = T2.id WHERE T2.name = "Kyle";``` |

(c) Case 3: Social Network Database Question

Figure 5: NL query from Spider and the corresponding results from different prompting approaches

and semantic similarity between user queries. These surface-level matching approaches often fail to identify the most effective examples because they consider only query semantics while ignoring the underlying SQL structural complexity.

One promising solution is to incorporate syntax structure information into the few-shot example selection process. This structure-aware approach would consider both semantic relevance and SQL structural patterns, enabling better matching of complex query requirements with appropriate examples.

### E.2 EXPLORING MORE CHALLENGING SCENARIOS

#### E.2.1 TEMPLATE-BASED SYNTHETIC DATA GENERATION FOR TEXT-TO-SQL TRAINING

Adapting a text-to-SQL model to a new database, like a company's proprietary database, requires developers to manually create extensive training data. This process requires: $(i)$ Writing natural language questions about the database. $(ii)$ Creating the corresponding correct SQL queries. $(iii)$ Validating the accuracy of both questions and SQL queries. This manual data collection process is not only time-consuming but also requires expertise in both SQL and the specific database domain, making it a significant bottleneck for the practical deployment of text-to-SQL systems.

Generating synthetic training data based on a template-based approach. This method aims to eliminate the need for manual data collection by systematically generating training examples using predefined syntax templates and database schema information. The generation process operates in three coordinated stages: template selection based on database schema, schema integration by populating templates with actual table and column names, and natural language query generation.

#### E.2.2 INTERACTION WITH DYNAMIC DATABASE

While current text-to-SQL methods, including our model, primarily focus on static databases, real-world databases are inherently dynamic. To develop a truly comprehensive database management system, it is essential to extend the Text-to-SQL framework to support full CRUD operations—Create, Read, Update, and Delete—enabling seamless and complete interaction with databases.

### E.3 BROADER APPLICATIONS

The structure-guided approach could be extended to other domains requiring structured output generation.

| Prompting strategy | PaLM-2 | CodeX | ChatGPT | GPT-4 |
|---|---|---|---|---|
| + Few-shot Prompting | 0.6985 | 0.7167 | 0.7394 | 0.7665 |
| + CoT Prompting | 0.6873 | 0.7198 | 0.7552 | 0.7834 |
| + SGU-SQL | 0.7395 | 0.7418 | 0.7846 | 0.8795 |

Table 8: Ablation Study: Performance comparison of different prompting strategies on the development set of Spider.

| Structure | Source Node x | Target Node y | Relation Type | Description |
|---|---|---|---|---|
| Query Structure | Question Concept | Question Concept | Forward-Syntax | y is the target word of x under syntax dependency. |
| | Question Concept | Question Concept | Backward-Syntax | y is the source word of x under syntax dependency. |
| | Question Concept | Question Concept | PartOf | x is a part or component of y under semantic parsing. |
| | Question Concept | Question Concept | Synonym | x is a synonym or equivalent to y under semantic parsing. |
| Database Structure | Column | Column | Foreign-Key | y is the foreign key of x. |
| | Table | Column | Has | The column y belongs to the table x. |
| | Table | Column | Primary-Key | The column y is the primary key of the table x. |
| Linking Structure | Question Concept | Table | None-Linking | No linking between x and y. |
| | Question Concept | Table | Partial-Linking | x is part of y, but the entire question does not contain y. |
| | Question Concept | Table | Exact-Linking | x and y are matched based on our Structure Linking model. |
| | Question Concept | Column | None-Linking | No linking between x and y. |
| | Question Concept | Column | Partial-Linking | x is part of y, but the entire question does not contain y. |
| | Question Concept | Column | Exact-Linking | x and y are matched based on our Structure Linking model. |
| | Question Concept | Column | Value-Linking | x is part of the candidate cell values of column y. |

Table 9: The relations used in three structures in SGU-SQL. All relations above are asymmetric.

### E.3.1 TEXT-TO-CYPHER (OTHER PROGRAMMING LANGUAGES)

Text-to-SQL converts natural language queries into SQL queries to interact with relational databases, while text-to-Cypher translates natural language into Cypher queries for graph database operations. Considering that data in graph databases is stored as nodes (entities) and edges (relationships) in the format of graphs, our SGU-SQL could be seamlessly applied on Text-to-Cypher.

### E.3.2 API PLANNING

API planning aims to generate a sequence of API calls to accomplish a given goal or user request. Each API is essentially a function with input parameters and return values. Each function can be treated as a table, where input parameters and return values are equivalent to columns in the table. Based on the data flow, we can build a graph to describe the dependencies between different APIs, transforming the API planning task into a problem similar to Text-to-SQL, as the dependency graph is analogous to the schema graph in text-to-SQL.

