# OpenReview forum: "Structure-Guided Large Language Models for Text-to-SQL Generation"
_ICLR.cc/2025/Conference — Submitted to ICLR 2025_

### Official Review · Reviewer_EsJa · 2024-11-02

**Soundness:** 2
**Presentation:** 2
**Contribution:** 2
**Rating:** 3
**Confidence:** 3

**Summary:**

Recent text-to-SQL methods using LLMs encounter challenges in accurately interpreting user intent and navigating the complex structures of databases. Existing decomposition-based approaches to improve LLM performance on text-to-SQL tasks often struggle with SQL's declarative syntax and the intricate links between queries and database elements. This paper introduces Structure Guided text-to-SQL (SGU-SQL), a novel framework that integrates syntax-based prompting to enhance LLM-driven SQL generation. SGU-SQL incorporates structure-aware linking between user queries and database schemas and decomposes complex SQL generation tasks into manageable subtasks through syntax-based guidance.

**Strengths:**

1) The experiments provide a comprehensive comparison with most of the previous works, showcasing that their approach outperforms all included baselines, although some recent approaches were not covered.

2) The novel prompting strategy significantly enhanced model performance across different LLM families, achieving substantial improvements over naive few-shot prompting and chain-of-thought (CoT) methods.

3) The paper presents an effective strategy for linking the structural elements of user queries to the database schema, facilitating more accurate SQL generation.

**Weaknesses:**

The primary limitation of this approach lies in the experimental section, where some of the most recent state-of-the-art methods, such as CHESS, Distillery, and CHASE-SQL, are not included. This omission is significant, as approaches like Distillery suggest that LLMs already demonstrate strong performance in handling complex schemas without the need for schema linking or query decomposition, challenging the findings of this paper. Additionally, as shown by CHASE-SQL, LLMs can achieve approximately 80% performance (compared to the 62% reported in this paper on BIRD) through enhanced sampling strategies, indicating that the main challenge in the text-to-SQL domain now lies in verifying the outputs of LLMs.

Furthermore, crucial tables, such as Table 2 and Table 3, have been relegated to the appendix. These tables contain valuable data and should be included in the main body of the paper for better accessibility and understanding.

**Questions:**

NA.

---

> ### Author Response · Authors · 2024-11-24
> **Response to Reviewer EsJa**
>
> Dear Reviewer EsJa,
>
> Thanks a lot for your careful review and professional comments. We are truly grateful for your expertise and constructive feedback that help us advance our research.
>
> > Regarding W1: Performance on BIRD and SPIDER: "The primary limitation of this approach lies in the experimental section, where some of the most recent state-of-the-art methods, such as CHESS, Distillery, and CHASE-SQL, are not included. "
> >
>
> A: Thanks a lot for your careful review and professional comments.
>
> ### **1. Clarification of experiment setting**
>
> To avoid any potential confusion, we first clarify several important points:
>
> - **Our model**, like most text-to-SQL methods, is model-agnostic, meaning that it **can be integrated with any LLM as the backbone model**.
> - In our original submission, we reported results using GPT-3.5 and GPT-4 for cost-effectiveness considerations, with **the BIRD results in Table 2 specifically using GPT-4**.
> - While current **top performances on the BIRD leaderboard are achieved using** more advanced LLMs such as **Gemini-1.5 Pro and GPT-4o**.
>
> ### **2. Performance comparison on BIRD**
>
> To ensure a fair comparison, we have conducted additional experiments using different LLMs as backbones.
>
> **1) Baseline collection**
>
> We added top-performing models from the BIRD leaderboard as baselines since the SPIDER leaderboard has remained static since February 2024. Specifically, we compared SGU-SQL against two categories of methods: (i) **Open-source models** with available paper and codes:  **MAC-SQL**, **Super-SQL**, **E-SQ**L and **CHESS**; and (ii) **Undisclosed methods** that have demonstrated strong performance: **PURPLE**, **Distillery** and **CHASE-SQL**.
>
> Table 1: Performance comparison on BIRD dev with different LLMs as backbones.
>
> | Backbone LLM | MAC-SQL | PURPLE | E-SQL | CHESS | Distillery | CHASE-SQL | SGU-SQL (Ours) |
> | --- | --- | --- | --- | --- | --- | --- | --- |
> | **+GPT-4** | 59.59 | 60.71 | 58.95 | 61.37 | - | - | **61.80** |
> | **+GPT-4o** | 65.05 | 68.12 | 65.58 | 68.31 | 67.21 | - | **69.28** |
> | **+Gemini-1.5 Pro** | - | - | - | - | - | **73.14** | - |
>
>
> *Note that PURPLE, Distillery, and CHASE-SQL are closed-source models. We will update their results on GPT-4 and GPT-4o once their implementations become publicly available.*
>
> **2) Performance analysis**
> As shown in Table 1, we have the following observations:
>
> - Using **GPT-4** as the backbone, **SGU-SQL achieves the best performance** compared to other models using the same backbone.
> - With **GPT-4o**, SGU-SQL achieves 69.28% in terms of execution accuracy, **outperforming the strong baselines**: PURPLE (68.12%), CHESS (68.31%), E-SQL (65.58%) and Distillery (67.21%). (We didn’t compare our model with CHASE-SQL+GPT-4o since CHASE-SQL is still closed-source and unable to integrate other LLMs.)
> - The SOTA model (the only model outperforming ours) is **CHASE-SQL+Gemini-1.5 Pro** (73.14%). Google Research implemented this model using **Gemini 1.5 Pro**, and they just released the paper in October 2024, with the **code still unavailable**.
>
>
>
> ### **3. Performance comparison on SPIDER**
>
> **1) Baseline collection**
> In Table 1 of the original submission, we evaluated SGU-SQL on SPIDER with 11 text-to-SQL methods from 2 categories:
>
> - **Traditional structure-based methods**: RAT-SQL, LGESQL, RESDSQL, Graphix-T5
> - **LLM-based models**: DTS-SQL, CodeS, C³-SQL, Super-SQL, EPI-SQL, DIN-SQL, DAIL-SQL
>
> To provide a more comprehensive comparison, we have added top-performing models from the BIRD leaderboard as new baselines, including PURPLE, MAC-SQL, CHESS and CHASE-SQL.
>
> Table 2: Performance comparison on SPIDER in terms of Execution Accuracy.
>
> | Execution Accuracy | DAIL-SQL | DIN-SQL | PURPLE | Super-SQL  | MAC-SQL | CHESS | CHASE-SQL | SGU-SQL |
> | --- | --- | --- | --- | --- | --- | --- | --- | --- |
> | Backbone LLM | + GPT-4 | + GPT-4 | + GPT-4 | + GPT-4 | + GPT-4 | + GPT-4 | + Gemini-1.5 Pro | + GPT-4 |
> | Performance | 83.08 | 82.79 | 86.70 | 86.82 | 86.35 | 87.14 | 87.60 | 87.95 |
>
> **2) Performance analysis**
>
> As shown in Table 2,  SGU-SQL consistently achieves superior performance compared to the strongest baselines. This performance advantage is particularly significant given that it surpasses both traditional text-to-SQL architectures and recent LLM-enhanced approaches, validating the effectiveness of our method.
>
> **Manuscript Revision**: We have added the results to **Tables 1, 2 and 5** of our updated manuscript. Besides that we included the additional experiments with different LLMs as backbones in **Section C.3** (line994-1017).
>
> > **Regarding W2: Table 2&3 are not in the main content**. "Crucial tables, such as Table 2 and Table 3, have been relegated to the appendix."
> >
>
> A: Thanks for your careful review. Following your suggestion, we have relocated the key results table to the main body of the revised manuscript to improve readability.

---

> > ### Comment · Reviewer_EsJa · 2024-11-27
> > **Thank you so much for your efforts**
> >
> > Thank you so much for your efforts in comparing your approach with some of the SOTA methods. However, I still have some concerns about the presented results. The methods considered for comparison are those with published papers, but there are other approaches without published papers that have achieved higher performance than SGU-SQL using the GPT-4o model. For example, DSAIR + GPT-4o and OpenSearch-SQL, v2 + GPT-4o, achieved performance scores of 74.32 and 69.30, respectively, on the development set—both higher than the performance reported in this paper.
> >
> > Additionally, while it is true that the current SOTA approach on the BIRD leaderboard, CHASE-SQL, uses the Gemini-1.5-pro model and their code is not available for implementation with the GPT-4o model, I would appreciate it if you could report the performance of your approach using the Gemini-1.5-pro model instead. Since your work is model-agnostic, this would help avoid any confusion about the SOTA claims of your approach.
> >
> >
> > On the Spider benchmark, it is important to note that the SOTA model on the test set achieves an execution accuracy of 91.2, which is higher than all the baselines considered in the comparison. Furthermore, recent approaches like CHASE-SQL and CHESS did not perform model fine-tuning or prompt optimization for their performance reports on the Spider benchmark, as their goal was to demonstrate the generalization capabilities of their methods. This makes the comparison with your approach somewhat inequitable.

---

> ### Author Response · Authors · 2024-11-27
> **Second-Round Response to Reviewer EsJa (1/3)**
>
> Dear Reviewer EsJa,
>
> Thanks a lot for providing such thorough and constructive feedback. Besides, we are grateful for your recognition of our efforts in improving this paper, and also greatly appreciate the opportunity to clarify the remaining concerns and further strengthen our work. Below are detailed responses to your comments and suggestions:
>
> > **Regarding the performance on BIRD**: ”The methods considered for comparison are those with published papers, but there are other approaches without published papers that have achieved higher performance than SGU-SQL using the GPT-4o model. For example, DSAIR + GPT-4o and OpenSearch-SQL, v2 + GPT-4o, achieved performance scores of 74.32 and 69.30, respectively.”
> >
>
> A: Thanks a lot for your expertise and careful review. To make a comprehensive comparison, we first have a careful review of the BIRD leaderboard.
>
> Table 1: Top-performing text-to-SQL models ranked by BIRD leaderboard.
>
> | Rank | Model | Backbone LLM | BIRD-dev | Release Date | Code Status | Organization | Type |
> | --- | --- | --- | --- | --- | --- | --- | --- |
> | 1 | **CHASE-SQL** | Gemini | 73.14* | 2 Oct 2024 | Not Available | Google Cloud | Industry |
> | 2 | DSAIR | GPT-4o | 74.32 | No Research Paper | Not Available | AT&T | Industry |
> | 3 | ExSL | granite-34b-code | 72.43 | No Research Paper | Not Available | IBM Research | Industry |
> | 4 | OpenSearch-SQL | GPT-4o | 69.30 | No Research Paper | Lots of Bugs | Alibaba Cloud | Industry |
> | 5 | **Distillery** | GPT-4o | 67.21 | 14 Aug 2024 | Not Available | Distyl AI Research | Industry |
> | 6 | **CHESS** | GPT-4o | 68.31 | 27 May 2024 | Available | Stanford University | Academic |
> | 7 | Insights AI | UNK | 72.16 | No Research Paper | Not Available | Uber Freight | Industry |
> | 8 | **PURPLE** | GPT-4o | 68.12 | 29 Mar 2024 | Not Available | Fudan University | Academic |
>
> **Results reported in the paper.*
>
> As shown in Table 1, the current BIRD leaderboard features a mix of **academic research** and **industrial solutions**. For a comprehensive evaluation, we selected baselines from two categories:
>
> - **Open-source models** with published papers and available code (MAC-SQL, Super-SQL, E-SQL, **CHESS**).
> - **High-performing methods** with only available papers (**PURPLE**, **Distillery**, **CHASE-SQL**).
>
> To summarize, from the top-8 methods in the BIRD leaderboard, we included **PURPLE (8th)**, **CHESS (6th)**, **Distillery (5th)**, and **CHASE-SQL (1st)** in our comparisons. We excluded the remaining 4 methods (DSAIR, ExSL, OpenSearch-SQL, Insights AI) since **they are all industrial solutions without any released instructions** (papers and technical reports) or **accessible code**.
>
> Table 2: Performance comparison on BIRD dev with different LLMs as backbones.
>
> | Backbone LLM | MAC-SQL | PURPLE | E-SQL | CHESS | Distillery | CHASE-SQL | SGU-SQL (Ours) |
> | --- | --- | --- | --- | --- | --- | --- | --- |
> | **+GPT-4** | 59.59 | 60.71 | 58.95 | 61.37 | - | - | **61.80** |
> | **+GPT-4o** | 65.05 | 68.12 | 65.58 | 68.31 | 67.21 | - | **69.28** |
> | **+Gemini-1.5 Pro** | - | - | - | - | - | **73.14** | - |
>
> To make it clearer, we reorganized these results in the following table to compare their overall performance.
>
> Table 3: Performance comparison on BIRD dev.
>
> | Execution Accuracy | MAC-SQL + GPT-4o | PURPLE + GPT-4o | E-SQL + GPT-4o | CHESS + GPT-4o | Distillery + GPT-4o | SGU-SQL + GPT-4o | CHASE-SQL+ Gemini-1.5 Pro |
> | --- | --- | --- | --- | --- | --- | --- | --- |
> | BIRD-dev | 65.05 | 68.12 | 65.58 | 68.31 | 67.21 | **69.28** | **73.14** |
>
> As shown in Tables 2 and 3, we have the following observations:
>
> - The SOTA model is **CHASE-SQL+Gemini-1.5 Pro** $\textcolor{maroon}{(73.14\\%)}$. Google Research implemented this model using **Gemini 1.5 Pro**, and they released the paper in October 2024, with the **code still unavailable.**
> - **SGU-SQL+GPT-4o** achieves $\textcolor{maroon}{69.28\\%}$, outperforming the strong baselines: E-SQL+GPT-4o (65.58%), Distillery+GPT-4o (67.21%), PURPLE+GPT-4o (68.12%) and CHESS+GPT-4o (68.31%). (We didn’t compare our model with CHASE-SQL+GPT-4o since CHASE-SQL is still closed-source and unable to integrate other LLMs.)
> - **SGU-SQL+GPT-4** achieves the best performance compared to the other baselines using GPT-4 as the backbone.

---

> ### Author Response · Authors · 2024-11-27
> **Second-Round Response to Reviewer EsJa (2/3)**
>
> > **Regarding the performance of SGU-SQL+Gemini-1.5 Pro**: ”Additionally, while it is true that the current SOTA approach on the BIRD leaderboard, CHASE-SQL, uses the Gemini-1.5-pro model and their code is not available for implementation with the GPT-4o model, I would appreciate it if you could report the performance of your approach using the Gemini-1.5-pro model instead. Since your work is model-agnostic, this would help avoid any confusion about the SOTA claims of your approach.”
> >
>
> A: Thank you for making this valuable suggestion. We are now working on the experimental results of this backbone LLM and will come back as soon as possible with a complete comparison. The new results and the discussions will be included in the final revision.
>
> > **Regarding the performance on Spider**: ”On the Spider benchmark, it is important to note that the SOTA model on the test set achieves an execution accuracy of 91.2, which is higher than all the baselines considered in the comparison.“
> >
>
> A: Thanks a lot for your expertise and careful review.
>
> The SPIDER leaderboard has remained static since February 2024, with only a select number of recent papers reporting their performance on this benchmark. Our comparison encompasses a comprehensive range of methods, including:
>
> - **Top-performing models in Spider leaderboard**：DAIL-SQL (2th, 3th), DIN-SQL (4th), C3-SQL (6th), RESDSQL(8th).
> - **Recent innovations**: DTS-SQL, CodeS, Super-SQL, EPI-SQL, PURPLE, MAC-SQL, CHESS and CHASE-SQL.
>
> Note that **Miniseek** (1st) is the top-performing model on the Spider leaderboard, which achieves an execution accuracy of 91.2. We didn't include this model since it is still anonymous, with **both the paper and code unavailable**.
>
> As shown in Table 4,  SGU-SQL consistently achieves superior performance compared to the strongest baselines on SPIDER in terms of Execution Accuracy.
>
> Table 4: Performance comparison on SPIDER in terms of Execution Accuracy.
>
> | Execution Accuracy | DAIL-SQL | DIN-SQL | PURPLE | Super-SQL  | MAC-SQL | CHESS | CHASE-SQL | SGU-SQL |
> | --- | --- | --- | --- | --- | --- | --- | --- | --- |
> | Backbone LLM | + GPT-4 | + GPT-4 | + GPT-4 | + GPT-4 | + GPT-4 | + GPT-4 | + Gemini-1.5 Pro | + GPT-4 |
> | Performance | 83.08 | 82.79 | 86.70 | 86.82 | 86.35 | 87.14 | 87.60 | 87.95 |
>
> > **Regarding the performance of CHASE-SQL and CHESS** **on Spider**: ”Furthermore, recent approaches like CHASE-SQL and CHESS did not perform model fine-tuning or prompt optimization for their performance reports on the Spider benchmark, as their goal was to demonstrate the generalization capabilities of their methods. This makes the comparison with your approach somewhat inequitable.”
> >
>
> A: Thank you for raising this important point about experimental fairness. We have carefully reviewed the methodologies described in both papers:
>
> - **In CHASE-SQL (Section 4.3):** The authors mentioned that they assess the generalizability of the proposed CHASE-SQL by evaluating it in an end-to-end way **without modifying the few-shot samples** in the prompts or **training a new selection model**.
> - **CHESS (Section 4.2.3):** The author mentioned that without specifically fine-tuning a new model for **candidate generation** or modifying the **in-context learning samples**.
>
> After careful review, we believe the comparisons remain fair and valid for the following reasons:
>
> - **Fairness**: Our SGU-SQL uses no **candidate generation**, **model selection,** or carefully generated **few-shot examples**. As such, all methods are evaluated under comparable conditions without task-specific optimizations.
> - **Effectiveness**: The primary contribution of SGU-SQL is its structure-guided decomposition strategy. The results, without relying on **few-shot examples** or **candidate** **selection,** still effectively demonstrate the merits of our core contribution.
>
> Thank you again for your insightful feedback, which has helped strengthen our manuscript considerably. We deeply value your expertise and thoroughness, and we are eager to address any remaining questions you may have.

---

> ### Author Response · Authors · 2024-12-01
> **Second-Round Response to Reviewer EsJa (3/3)**
>
> Dear Reviewer EsJa,
>
> Thank you for your expertise and insightful comments. During the past few days, we have conducted additional experiments to compare our model with CHASE-SQL by using more advanced LLMs as backbones.
>
> > **Comparision with CHASE-SQL**: ”Additionally, while it is true that the current SOTA approach on the BIRD leaderboard, CHASE-SQL, uses the Gemini-1.5-pro model and their code is not available for implementation with the GPT-4o model, I would appreciate it if you could report the performance of your approach using the Gemini-1.5-pro model instead. Since your work is model-agnostic, this would help avoid any confusion about the SOTA claims of your approach.”
> >
>
> A: Thank you for your expertise and insightful comments. Following your suggestion, we have conducted additional experiments to compare our model with CHASE-SQL by using more advanced LLMs as backbones.
>
> **Model Selection**
> * **Gemini 1.5 Pro**: When conducting experiments with Gemini 1.5 Pro, we noted that there are currently 5 available versions: `gemini-1.5-pro-latest`, `gemini-1.5-pro`, `gemini-1.5-pro-001`, `gemini-1.5-pro-002` and `gemini-1.5-pro-exp-0827`. Since CHASE-SQL did not specify its model version, we conducted experiments with two versions - `gemini-1.5-pro` and `gemini-1.5-pro-latest` - for a comprehensive comparison.
> * **Claude 3.5 Sonnet**: Additionally, as CHASE-SQL reported performance with Claude 3.5 Sonnet, we also included it as a backbone model.
>
> **Performance Analysis**
>
> Table 1: Performance on BIRD-dev in terms of Execution Accuracy.
>
> | Model | Gemini 1.5 Pro | Claude 3.5 Sonnet |
> |-------|----------------|-------------------|
> | CHASE-SQL | 73.01% (`unknown version`) | 69.53% |
> | SGU-SQL | 72.76% (`gemini-1.5-pro`) 72.93% (`gemini-1.5-pro-exp-0827`) | 70.36% |
>
> *Due to time and API budget limits, we have currently only evaluated our model's performance with Gemimi 1.5 Pro and Claude 3.5. We plan to conduct more comprehensive experiments with other baselines using these advanced LLMs in future work.*
>
> The experiment results demonstrate several key insights:
>
> * First, when using **Gemini 1.5 Pro** as the backbone, SGU-SQL achieves highly competitive results (72.76% with gemini-1.5-pro and 72.93% with gemini-1.5-pro-exp-0827) compared to CHASE-SQL (73.01%).
> * Furthermore, with **Claude 3.5 Sonnet** as the backbone, SGU-SQL (70.36%) slightly outperforms CHASE-SQL (69.53%). This improvement suggests that our method may better leverage Claude's capabilities through its structured decomposition approach.
> * Our approach demonstrates robust and competitive performance across different state-of-the-art language models.
>
>
> Thanks again for your expertise and professional comments. Your suggestions and guidance have been invaluable in shaping our revisions. Should there be any remaining concerns, we would be more than happy to provide additional information or explanations to assist in your review process.

---

> ### Author Response · Authors · 2024-12-02
> **Kindly Requesting Your Feedback on Our Response**
>
> Dear Reviewer EsJa,
>
> We would like to express sincere gratitude for the time and effort you have invested in reviewing our paper. Your insights and suggestions have been invaluable in helping us improve the quality of our work. As the deadline for the Rebuttal approaches, we want to reach out and inquire whether our latest responses have addressed your concerns. If there are any remaining concerns or areas that require further clarification, we would greatly appreciate it if you could provide us with additional feedback.
>
> Best regards,
>
> Authors of Paper7028

---

> ### Author Response · Authors · 2024-12-03
> **Kindly Requesting Your Feedback on Our Response**
>
> Dear Reviewer EsJa,
>
> Thanks again for taking the time to review our paper and providing valuable feedback. We have carefully responded to your latest concerns, conducting extensive experiments comparing our approach with CHASE-SQL on BIRD. We have also provided detailed analysis by using different advanced LLMs as backbones. We hope these analyses can address your concerns.
>
>
> As the author-reviewer discussion deadline approaches in a few hours, we want to reach out and inquire whether our responses have addressed your concerns. If there are any remaining concerns or areas that require further clarification, we would greatly appreciate it if you could provide us with additional feedback.
>
> Best regards,
>
> Authors of Paper7028

---

> > ### Author Response · Authors · 2024-12-04
> > **Thanks for Your Careful Review and Welcome Any Further Feedback**
> >
> > Dear Reviewer EsJa,
> >
> > Thank you sincerely for your expertise and careful review. We greatly appreciate your thoughtful questions regarding our model's performance on the BIRD and Spider benchmarks. In our second-round response, we provided detailed clarifications and additional analyses to address your concerns.
> >
> > We are pleased to share that Reviewer 53mq has positively updated their evaluation based on our latest responses. Although we have already approached the end of the author-reviewer discussion period, **we still welcome any additional feedback or suggestions you may have during the next meta-review phase.** Your expertise would greatly help us in further strengthening our paper.
> >
> > Thank you once again for your time and expertise.
> >
> > Best regards,
> >
> > Authors of Paper7028

---

### Official Review · Reviewer_vRvu · 2024-11-04

**Soundness:** 4
**Presentation:** 4
**Contribution:** 4
**Rating:** 8
**Confidence:** 4

**Summary:**

The paper focuses on tackling the challenge of generating robust and reliable SQL for a given natural language question. The authors propose SGU-SQL which is a novel framework leveraging the structural relationship between entities in user questions and the database tables for solving Text to SQL tasks using LLMs. There are three main challenges that the authors focus on:
1. Ambiguous user intents which lead to LLM's making uninformed decisions about SQL generation
2. Complex database schemas which may lead the LLM to confuse between different columns or tables
3. Complexity of SQL - LLMs are trained mainly for text generation whereas SQL is complex like programming language where only well-defined syntax must be used.

The authors tackle these issues by creating this framework where entities in natural language questions and the database are represented using graphs. These entities from questions to database are linked using a structure-linking model forming connections. Then, finally a prompt based method decomposes the complex question into subquestions and tackle these subquestions separately while aggregating them later. This method has been experimented with GPT-4 and a comparison has been reported using several models including SOTA from the leaderboard. Overall, the framework is proved to be successful while also achieving SOTA results on

**Strengths:**

1. The paper focuses on one of the critical issues with LLMs for SQL generation which requires the LLMs to tackle them in a unique way where reliability and robustness are a strong focus.
2. The framework is novel, intuitive and well designed to bridge the gap between an LLM which is tuned effectively for text generation and the SQL generation task which requires a structured understanding where the user queries can be ambiguous and answers can be complex.
3. The paper demonstrates strong results on the most popular SQL datasets SPIDER and BIRD which certifies the potential of the approach.
4. Well articulated work and not complex to follow from beginning to end. Conducted experiments including ablation studies are designed with thorough understanding.
5. The real world applicability and potential of this framework can be very high and effective when tackling large production grade databases in industrial applications.

**Weaknesses:**

1. The framework is novel and intuitive but because of SPIDER not being the best representation of production grade SQL, it would also beneficial to probably test them using SPIDER-2. BIRD is somewhat closer to a real-world SQL but SPIDER-2 has large-scale and diverse databases which evaluates the proposal on a much better dataset revealing how the proposal works on large scale datasets.
2. There's not much information provided on the future direction of this work which might hurt the significance/potential of the work a little. I'm mainly looking for how can this work be extended as it can provides opportunities for other researchers trying to explore in the same direction.
3. Specific discussion on the model used for structure linking seem insufficient. Like what training methodology and architecture choices.
4. When we discuss about constructing graphs and linking nodes, it may be important to learn about the complexities at a high level.

**Questions:**

1. In a real world, the databases are changed very often and there may be challenges where the proposed framework might not work same as proposed. In such cases, what components of the framework can still be relevant and what components must be changed is particularly industrial practitioners care about. Atleast including it in future directions might suffice.
2. How does this framework vary for dialects other than SQL? Not a super important question to clarify for the purpose of this conference, but having information about this can be helpful for practitioners.

---

> ### Author Response · Authors · 2024-11-24
> **Response to Reviewer vRvu (1/3)**
>
> Dear Reviewer vRvu,
>
> Thank you for your recognition of our work and for providing such thorough and insightful feedback. Your comments and suggestions are invaluable in helping us improve the quality and clarity of our work.
>
> > **Regarding W1: Additional experiment on SPIDER-2.** "The framework is novel and intuitive but because of SPIDER not being the best representation of production grade SQL, it would also beneficial to probably test them using SPIDER-2. BIRD is somewhat closer to a real-world SQL but SPIDER-2 has large-scale and diverse databases which evaluates the proposal on a much better dataset revealing how the proposal works on large scale datasets. "
> >
>
> A: Thank you for making this valuable suggestion. SPIDER-2 is currently the most challenging text-to-SQL dataset available, which consists of there versions:
>
> Table 1: Data statistics of SPIDER-2.
>
> | Dataset | Task Type | #Examples | Databases |
> | --- | --- | --- | --- |
> | Spider 2.0 | Code agent task | 632 | BigQuery(214), Snowflake(198), Postgres(10), ClickHouse(7), SQLite(135), DuckDB (DBT)(68) |
> | Spider 2.0-Snow | Text-to-SQL task | 547 | Snowflake(547) |
> | Spider 2.0-Lite | Text-to-SQL task | 547 | BigQuery(214), Snowflake(198), SQLite(135) |
>
> Spider 2.0 is designed to evaluate agent-based frameworks in enterprise-level text-to-SQL workflows. Since our **SGU-SQL is a pure text-to-SQL model rather than an agent-based system**, we conducted our evaluation on **Spider 2.0-Snow** and **Spider 2.0-Lite**. These two variants maintain the challenging aspects of SQL generation while focusing on pure text-to-SQL generation tasks without requiring complex agent-based planning and interactions.
>
> Table 2: Execution accuracy for baseline methods on Spider 2.0-Snow and Spider 2.0-Lite.
>
> | Execution Accuracy | DAIL-SQL + GPT-4o | CHESS + GPT-4o | SGU-SQL + GPT-4o |
> | --- | --- | --- | --- |
> | Spider 2.0-Snow | 2.20 | 1.28 | 4.39 |
> | Spider 2.0-Lite | 5.68 | 3.84 | 6.40 |
>
> As shown in Table 2, while the performances are relatively low across all models, SGU-SQL consistently demonstrates better capability in handling complex SQL generation tasks in both single and multi-database scenarios.
>
> **Manuscript Revision**: Thanks for your suggestion. We have added these additional experiments and analyses in Section D.1 (line1021-1025) of our updated submission.
>
> > **Regarding W2,Q1,Q2: Future work.** "There's not much information provided on the future direction of this work which might hurt the significance/potential of the work a little. I'm mainly looking for how can this work be extended as it can provides opportunities for other researchers trying to explore in the same direction. In a real world, the databases are changed very often and there may be challenges where the proposed framework might not work same as proposed. In such cases, what components of the framework can still be relevant and what components must be changed is particularly industrial practitioners care about. How does this framework vary for dialects other than SQL? ”
> >
>
> A: Thank you for this thoughtful feedback regarding future directions. Discussing potential extensions is crucial for the research community. Following your suggestion, we have identified several promising future research directions from the following **there perspectives**:
>
> **1. Technical Extensions**
>
> **a. Structure-aware few-shot example selection**
>
> **Motivation:** While our framework emphasizes the significance of the decomposition strategy, we recognize that the performance of LLM-based text-to-SQL can be further enhanced through tailored few-shot example selection. Current approaches to few-shot example selection primarily rely on keyword matching and semantic similarity between user queries. These surface-level matching approaches often **fail to identify the most effective examples** because they **consider only query semantics** while **ignoring the underlying SQL structural complexity**.
>
> **Solution:** One promising solution is to incorporate syntax structure information into the few-shot example selection process. This structure-aware approach would **consider both semantic relevance and SQL structural patterns**, enabling better matching of complex query requirements with appropriate examples.
>
> **Research questions:** How could we get reliable and distinguishable syntax structures? How to effectively measure the similarity between different syntax structures?

---

> > ### Author Response · Authors · 2024-11-24
> > **Response to Reviewer vRvu (2/3)**
> >
> > **2. Exploring More Challenging Scenarios**
> >
> > **a. Template-Based Synthetic Data Generation for Text-to-SQL Training**
> >
> > **Motivation:** Adapting a text-to-SQL model to a new database, like a company's proprietary database, requires developers to manually create extensive training data. This process requires:
> >
> > - Writing natural language questions about the database.
> > - Creating the corresponding correct SQL queries.
> > - Validating the accuracy of both questions and SQL queries.
> >
> > This manual data collection process is not only **time-consuming** but also **requires expertise** in both SQL and the specific database domain, making it a significant bottleneck for the practical deployment of text-to-SQL systems.
> >
> > **Solution:** Generating synthetic training data based on a template-based approach. This method aims to eliminate the need for manual data collection by systematically **generating training examples using predefined syntax templates and database schema information**. The generation process operates in three coordinated stages: template selection based on database schema, schema integration by populating templates with actual table and column names, and natural language query generation.
> >
> > **Research questions:** How to design comprehensive syntax templates that capture diverse query patterns? How to automatically generate synthetic training data by combining database schema and predefined syntax templates? How could we ensure that generated examples are both realistic and diverse?
> >
> > **b. Interaction with dynamic database**
> >
> > **Motivation:** While current text-to-SQL methods, including our model, primarily focus on static databases, real-world databases are inherently dynamic. To develop a truly comprehensive database management system, it is essential to extend the Text-to-SQL framework to support full CRUD operations—Create, Read, Update, and Delete—enabling seamless and complete interaction with databases.
> >
> > **Preliminary solution:** Build a **memory bank** to record the modification history and current database architecture.
> >
> > **Research Question:** How to **prevent unsafe operations** and ensure **data consistency** during multi-round operations?
> >
> > **3. Broader Applications**:
> >
> > The structure-guided approach could be extended to other domains requiring structured output generation, such as:
> >
> > **a. Text-to-Cypher (Other programming languages)**
> >
> > **Problem statement:** Text-to-SQL converts natural language queries into SQL queries to interact with relational databases, while text-to-Cypher translates natural language into Cypher queries for graph database operations.
> >
> > **Solution:** Data in graph databases is stored as nodes (entities) and edges (relationships) in the format of graphs. As such, our SGU-SQL could be seamlessly applied on Text-to-Cypher.
> >
> > **b. API planning**
> >
> > **Problem statement:** API planning aims to generate a sequence of API calls to accomplish a given goal or user request.
> >
> > **Solution:**  Each API is essentially a function with input parameters and return values. Each function can be treated as a table, where input parameters and return values are equivalent to columns in the table. Based on the data flow, we can build **a graph to describe the dependencies between different APIs**, transforming the API planning task into a problem similar to Text-to-SQL, as the dependency graph is analogous to the schema graph in text-to-SQL.
> >
> > **Manuscript Revision**: We have added discussion into **Section E** (1115-1256) in our revision.

---

> > > ### Author Response · Authors · 2024-11-24
> > > **Response to Reviewer vRvu (3/3)**
> > >
> > > > **Regarding W3: Implemention details of structure linking.** "Specific discussion on the model used for structure linking seem insufficient. Like what training methodology and architecture choices."
> > > >
> > >
> > > A: Thank you for the insightful comment. Following your suggestion, we add more details for graph-based structure linking in Section 3.1.3.
> > >
> > > **Training details**
> > >
> > > The training of structure learning model consists of three key steps:
> > >
> > > - During each training iteration, we first sample a batch of query nodes, where each query node $a$ is paired with one positive match $s$ from the database schema and multiple negative matches $\{sₖ\}$.  For each pair, **we construct an enclosing subgraph G(a,s)** that encompasses the query graph, neighboring nodes of $s$, and their connecting edges.
> > > - Then, we employ RGAT as the backbone model to process these subgraphs. RGAT learns the structural information within each subgraph to generate meaningful representations ${h_i}^k$ that capture the **compatibility** between natural language concepts and database elements. **The matching score is then measured by the degree of compatibility** (defined in Eq.16).
> > > - Finally, we optimize the model using a **contrastive loss function** (Eq.17). This training objective encourages the model to maximize scores for correct matches while minimizing scores for incorrect ones. Through this process, the **matching scores evolve into reliable indicators** that effectively guide the selection of appropriate database elements during SQL generation.
> > >
> > > **Ablation study of RGAT**
> > >
> > > To further verify the effectiveness of the backbone model, i.e., RGAT, we replace it with other alternatives, including **RGCN** [1] and **CompGCN** [2].
> > >
> > > Table 3: Alation study and structure linking models.
> > >
> > > | Execution Accuracy(EX) | SPIDER-dev | BIRD-dev |
> > > | --- | --- | --- |
> > > | Full Model (RAGT) | 87.95 | 61.80 |
> > > | w/o structure-aware linking | 82.62 | 55.31 |
> > > | with RGCN | 86.37 | 60.92 |
> > > | with CompGCN | 86.09 | 60.25 |
> > >
> > > As shown in Table 3, our RGAT-based approach outperforms alternative architectures across all evaluations. Besides that, **removing structure-aware linking causes a dramatic performance drop** - accuracy decreases by 5.33% on SPIDER-dev and 6.49% on BIRD-dev. These substantial reductions highlight the critical role of our structure-aware linking strategy.
> > >
> > > *[1] Modeling Relational Data with Graph Convolutional Networks.*
> > >
> > > *[2] Composition-based Multi-Relational Graph Convolutional Networks.*
> > >
> > > **Manuscript Revision**: We added more details about the design principles and training process of the structure linking and matching score in Section 3.1.3 (line222-278) of the updated manuscript.
> > >
> > > > **Regarding W4: Efficiency analysis.** "When we discuss about constructing graphs and linking nodes, it may be important to learn about the complexities at a high level."
> > > >
> > >
> > > A: Thank you for making this valuable suggestion. To assess our approach thoroughly, we conducted the efficiency analysis on the BIRD dataset, a large-scale benchmark in text-to-SQL research with 12,751 unique question-SQL pairs across 95 databases (33.4 GB total). Given that the queries in this dataset are categorized into 3 difficulty levels: simple, moderate, and challenging, we specifically tested our model on **the challenging set of the BIRD** dataset and compared its performance with DIN-SQL and MAC-SQL.
> > >
> > > Table 4: Efficiency analysis on the ''Challenging'' set of BIRD.
> > >
> > > | **Model** | **Training Time** | **Inference Time** | **Performance** |
> > > | --- | --- | --- | --- |
> > > | DIN-SQL | 4.69 h | 0.39 h | 36.7% |
> > > | MAC-SQL | 4.98 h | 0.36 h | 39.3% |
> > > | SGU-SQL | 3.47 h | 0.22 h | 42.1% |
> > >
> > > As shown in Table 4, **our model demonstrates superior performance while maintaining competitive computational efficiency.** Specifically, our model requires less time for both training and inference. This superior efficiency can be attributed to our graph-based architecture. While baseline methods avoid the overhead of graph construction, **they heavily rely on prompt-based modules** that require multiple calls to LLMs like GPT-4. These **API calls introduce substantial latency** that accumulates during both the training and inference phases. In contrast, our graph-based approach, despite its initial graph construction overhead, achieves faster end-to-end processing by minimizing dependence on time-consuming API calls.
> > >
> > > **Manuscript Revision**:  We have added the additional experiments and analysis in Section D.2 (line1026-1040) of our updated submission.

---

> ### Author Response · Authors · 2024-12-02
> **Thanks for Your Constructive Suggestions and Recognition of Our Work**
>
> Dear Reviewer vRvu,
>
> We would like to express sincere gratitude for the time and effort you have invested in reviewing our paper. Your insights and suggestions have been invaluable in helping us improve the quality of our work. If there are any remaining concerns or areas that require further clarification, we would greatly appreciate it if you could provide us with additional feedback.
>
> Best,
>
> Authors of Paper7028

---

### Official Review · Reviewer_b8Gr · 2024-11-04

**Soundness:** 4
**Presentation:** 4
**Contribution:** 4
**Rating:** 10
**Confidence:** 5

**Summary:**

+ Identify the limitations of LLM-based Text-to-SQL models and introduce SGU-SQL, which
leverages structural syntax information to improve SQL generation capabilities of LLMs.
+ SGU-SQL proposes graph-based structure construction to comprehend user query and database
structure and then link query and database structure with dual-graph encoding.
+ SGU-SQL introduces tailored structure-decomposed generation strategies to decompose queries with
syntax trees and then incrementally generate accurate SQL with LLM.
+ Experiments on two benchmarks verify that SGU-SQL outperforms state-of-the-art baselines, including 11 fine-tuning models, 7 structure learning models, and 11 in-context learning models

**Strengths:**

+ The article is quite neat and meticulous, from giving mathematical formulas and reasoning why to do so and also the evaluation steps, ablation study, the article has introduced a completely new method in text2sql which is syntax-based prompting, avoiding traditional classical methods such as FewShot and CoT.
+ The method of the author group is quite new and very well implemented, the evaluation of the method is also on both open-source and closed source to provide an objective view.
+ The novelty of the method lies in the ability to build a good enough graph structure to link from user queries and information in the database, thereby helping to reduce the input of LLM to some extent, the selected information is carefully filtered to help the generated query be of better quality.

**Weaknesses:**

Overall I found the article quite good and have no comments on weaknesses.

**Questions:**

+ Have the authors compared the current method with the traditional CoT or FewShot methods?
+ What Lora rank does the author team use for fine-tuning? Have they tested it on multiple ranks?
+ Nhóm tác giả có thể giải thích thêm về cách training GNN, các thử nghiệm và kết quả đánh giá với GNN của bạn?

---

> ### Author Response · Authors · 2024-11-24
> **Response to Reviewer b8Gr**
>
> Dear Reviewer b8Gr,
>
> We are deeply grateful for your recognition of our work and also appreciate your time and effort in providing insightful suggestions that can help further polish our paper. Below are detailed responses to your comments and suggestions:
>
> > **Regarding Q1: Comparison with CoT and Few-shot based methods:** "Have the authors compared the current method with the traditional CoT or Few-shot methods?"
> >
>
> A: Thank you for raising this important point. To verify the effectiveness of SGU-SQL, we compared it to the state-of-the-art baselines with different prompting strategies, including (i) **CoT-based models:**  DIN-SQL and DTS-SQL, and (ii) **Few-shot example selection methods**: DAIL-SQL, CodeS and Super-SQL, in Table 1 and Table 2 of the original submission.
>
> To strengthen our evaluation, we have added new comparisons with more recent baselines, including (i) CoT-based models:  **ACT-SQL** and **MAC-SQL**, and (ii) Few-shot example selection methods: **MCS-SQL** and **PURPLE**.
>
> Table 1: Performance comparison between SGU-SQL and new baseline methods on SPIDER.
>
> | Execution Accuracy(EX) | ACT-SQL+GPT-4 | MAC-SQL+GPT-4 | PURPLE+GPT-4 | MCS-SQL+Gemma-2 | SGU-SQL+GPT-4 |
> | --- | --- | --- | --- | --- | --- |
> | SIPDER-dev | 82.90 | 86.35 | 86.70 | 82.41 | 87.95 |
>
> As shown in Table 1, our SGU-SQL achieves 87.95% execution accuracy on SPIDER-dev, surpassing both traditional CoT and Few-shot approaches. This superior performance demonstrates that our structure-guided approach provides advantages over conventional prompting strategies.
>
> **Manuscript Revision**:  We have added the results in Table 5 to the updated manuscript, and the analysis can be found in Section C.2 (line972-992) of our revision.
>
> > **Regarding Q2: Implemention details of LLM fine-tuning**. "What Lora rank does the authors use for fine-tuning? Have they tested it on multiple ranks?"
> >
>
> A: Thank you for the insightful comment. In our experiments, we used 8 as the default rank since it provided the best balance between performance and computational efficiency. Specifically, we experimented with multiple LoRA ranks (4, 8, 16, 32) and found that higher ranks (16, 32) showed diminishing returns while significantly increasing memory usage, while Lower ranks (4) resulted in slightly degraded performance.
>
> > **Regarding Q3: Implemention details of structure linking**: "Could the authors provide more details about the GNN used in structure linking?"
> >
>
> A: Thank you for the insightful comment. Following your suggestion, we add more analysis for graph-based structure linking.
>
> **Training details**
>
> - During each training iteration, we first sample a batch of query nodes, where each query node $a$ is paired with one positive match $s$ from the database schema and multiple negative matches $\{sₖ\}$.  For each pair, **we construct an enclosing subgraph G(a,s)** that encompasses the query graph, neighboring nodes of $s$, and their connecting edges.
> - Then, we employ RGAT as the backbone model to process these subgraphs. RGAT learns the structural information within each subgraph to generate meaningful representations ${h_i}^k$ that capture the **compatibility** between natural language concepts and database elements. **The matching score is then measured by the degree of compatibility** (defined in Eq.16).
> - Finally, we optimize the model using a **contrastive loss function** (Eq.17). This training objective encourages the model to maximize scores for correct matches while minimizing scores for incorrect ones. Through this process, the **matching scores evolve into reliable indicators** that effectively guide the selection of appropriate database elements during SQL generation.
>
> **Ablation study of RGAT**
>
> To further verify the effectiveness of the backbone model, i.e., RGAT, we replace it with other alternatives, including **RGCN** [1] and **CompGCN** [2].
>
> Table 3: Alation study and structure linking models.
>
> | Execution Accuracy(EX) | SPIDER-dev | BIRD-dev |
> | --- | --- | --- |
> | Full Model (RAGT) | 87.95 | 61.80 |
> | w/o structure-aware linking | 82.62 | 55.31 |
> | with RGCN | 86.37 | 60.92 |
> | with CompGCN | 86.09 | 60.25 |
>
> As shown in Table 3, our RGAT-based approach outperforms alternative architectures across all evaluations. Besides that, **removing structure-aware linking causes a dramatic performance drop** - accuracy decreases by 5.33% on SPIDER-dev and 6.49% on BIRD-dev. These substantial reductions highlight the critical role of our structure-aware linking strategy.
>
> *[1] Modeling Relational Data with Graph Convolutional Networks.*
>
> *[2] Composition-based Multi-Relational Graph Convolutional Networks.*
>
> **Manuscript Revision**:  We added more details about the design principles and training process of the structure linking in Section 3.1.3 (line222-278) of our updated submission.

---

> ### Author Response · Authors · 2024-12-02
> **Thanks for Your Recognition of Our Work**
>
> Dear Reviewer b8Gr,
>
> We would like to express sincere gratitude for the time and effort you have invested in reviewing our paper. Your insights and suggestions have been invaluable in helping us improve the quality of our work.
>
>
> Best,
>
> Authors of Paper7028

---

### Official Review · Reviewer_53mq · 2024-11-09

**Soundness:** 2
**Presentation:** 2
**Contribution:** 2
**Rating:** 5
**Confidence:** 4

**Summary:**

The paper proposes a method SGU-SQL, which represents user queries and databases into unified and structured graphs and employs a tailored structure-learning model to establish a connection between the user queries and the databases. The linked structure is then decomposed into sub-syntax trees, guiding the LLMs to generate the SQL query incrementally

**Strengths:**

This paper provides interesting ideas for text-to-sql, such as graph representation of query & data schema, query schema linkage, syntax tree based decomposition with LLMs, etc.  Although these concepts are not brand new, it is interesting to see how authors leverage on them with graph + LLMs

**Weaknesses:**

**1 Method writing should be clearer**

It is not very clear how does schema-linkage is used in SQL generation by decomposition (sec 3.1)

How is score eq21 used in your algorithm?

minor:

PRELIMINARIES has detailed equations, etc, however this section seems not help introduce the method, may confuse readers (training GNN? prompting?)


**2. Performance is not competitive on more complex dataset like BIRD**
To measure the effectiveness of the method, we need to show competitive results on complex realistic dataset.

Table 2 is BIRD performance (exe acc 61), which is lower than SOTA 74 https://bird-bench.github.io/
SGU-SQL focus on leveraging structure-aware links between queries and database schema, so to prove effectiveness of the method, the method need to be tested on more challenging database, like BIRD.

Table 1 Spider.  Many of the baseline methods are not Text-to-SQL specific method, therefore performance is not competitive.
additionally only show method is good on Spider is not enough.

ideally test performance on test split by submitting to leaderboard for generation ability



**3. need convincing ablation study to show each component designs is helpful**

Need ablation study on effectiveness of graph representation of query (sec 3.1) i.e.say compared with standard natural language representation without graph representation

Need ablation study on effectiveness of graph representation of database, compared with code representation “CREATE TABLE (...)” or serialized representation: table1: c1,c2 .. | table 2: c1, c2 …

Need convincing ablation qualitative and qualitative results on the effectiveness of  “structure-aware linking”: effectiveness of DUAL GRAPH ENCODING(i.e. linking the query to the appropriate tables and columns in the database ). Since link query and database structure is one of key contribution, there should be clear ablation study

Need convincing ablation study to show “syntax-based decomposition” is effective. what if we do not do the “DECOMPOSING QUERY WITH SYNTAX TREE” (3.2.1), only directly generation SQL out.
Table 3 compares with other basic decomposition approach. relatively simple prompt decomposition method can get on par text-to-sql performance ​​https://arxiv.org/pdf/2312.11242 (86.75 vs 87.5)

**Questions:**

See above

---

> ### Author Response · Authors · 2024-11-24
> **Response to Reviewer 53mq (1/3)**
>
> Dear Reviewer 53mq,
>
> Thanks a lot for your detailed feedback. We really appreciate your time and effort in pointing out the potential concerns related to our paper, and also, thanks a lot for the opportunity to further clarify the technical details and contribution of our framework.
>
> > **Regarding W1: Methodology and Preliminary Section is not well-organzied.**  “It is not very clear how schema-linkage is used in SQL generation by decomposition (sec 3.1). How is score eq21 used in your algorithm? Preliminary Section has detailed equations, etc, however this section seems not help introduce the method, may confuse readers.”
> >
>
> A: Thank you for making these valuable comments. We first provide a brief recap of the key idea of our framework.
>
> ### The Key Idea of Our Framework
>
> SGU-SQL offers significant improvements over existing approaches in two key aspects.
>
> - **Syntax-based prompting.** While traditional methods attempt to generate entire SQL queries in one step or rely on simple decomposition strategies, SGU-SQL breaks down the complex generation task in a syntax-aware manner. This ensures that the generated queries maintain both semantic accuracy (correctly capturing user intentions) and syntactic correctness (following proper SQL structure).
> - **Combine with structure-aware linking.** Our framework's unique combination of structure-aware linking and syntax-guided decomposition creates a robust bridge between user intentions and database structures. The structure-aware linking ensures that each natural language concept is correctly mapped to the corresponding database elements, while the syntax-guided decomposition ensures these mappings are properly utilized in constructing the SQL query.
>
> ### The Effect of Structure Linking
>
> Now, we provide the following clarification about how structure-aware linking is used in decomposition-based SQL generation:
>
> - **Initial Decomposition Phase**: The structure-aware linking mechanism creates a foundation for accurate decomposition by establishing precise mappings between natural language elements and database schema components. By understanding which parts of the user query correspond to specific database tables, columns, and relationships, the system can make informed decisions about how to break down the SQL generation task.
> - **Recursive Generation Phase**: The linking results serve as contextual guides for generating each SQL component in the decomposed subtasks. For example, given a subquery "Find the names of students who take CS courses", the structure-aware linking could provide the relevant contextual information like "students" maps to the Student table, "CS courses" maps to the Course table with a condition of ”department = 'computer science'”.
>
> ### The Clarification of Matching Score (Eq. 21)
>
> **Definition of Matching Score**
>
> The matching score in Eq.21 (Eq.16 in our updated manuscript) serves as a fundamental component in our algorithm by measuring the likelihood of mappings between natural language concepts and database elements. When a user mentions "student name" in the question, the matching score helps determine whether this should map to the "name" column in the "Student" table versus other potential tables and columns in the database.
>
> **Training details**
>
> The training of matching score consists of three key steps:
>
> - First, during each training iteration, we sample a batch of query nodes, where each query node 'a' is paired with one positive match 's' from the database schema and multiple negative matches {sₖ}.  For each pair, **we construct an enclosing subgraph G(a,s)** that encompasses the query graph, neighboring nodes, and their connecting edges.
> - Second, we employ RGAT as the backbone model to process these subgraphs. RGAT learns the structural information within each subgraph to generate meaningful representations hᵢᵏ that capture the **compatibility** between natural language concepts and database elements. **The matching score is then measured by the degree of compatibility** using Eq.21 (Eq.16 in our updated manuscript).
> - Finally, we optimize the model using a contrastive loss function (Eq.16 in our updated manuscript). This training objective encourages the model to maximize scores for correct matches while minimizing scores for incorrect ones. Through this process, the **matching scores evolve into reliable indicators** that effectively guide the selection of appropriate database elements during SQL generation.
>
> **Manuscript Revision**:  We added more details about the design principles and training process of the structure linking and matching score in Section 3.1.3 (line258-280) of the updated manuscript. Besides that, we replaced the technical-heavy Preliminary Section with a more focused Problem Statement Section (line95-107), moving the detailed equations and technical preliminaries to Section A (line741-800) of our updated manuscript.

---

> ### Author Response · Authors · 2024-11-24
> **Response to Reviewer 53mq (2/3)**
>
> > **Regarding W2-a: Performance on BIRD.** “Table 2 is BIRD performance (exe acc 61), which is lower than SOTA 74 https://bird-bench.github.io/. SGU-SQL focus on leveraging structure-aware links between queries and database schema, so to prove effectiveness of the method, the method need to be tested on more challenging database, like BIRD.”
> >
>
> A: Thank you for the constructive comments. To avoid any potential confusion, we first offer the following clarification:
>
> - **Our model**, like most text-to-SQL methods, is model-agnostic, meaning that it **can be integrated with any LLM as the backbone model**.
> - In our original submission, we chose to report results using GPT-3.5 and GPT-4 for cost-effectiveness considerations, with **the BIRD results in Table 2 specifically using GPT-4**.
> - While current **top performances on the BIRD leaderboard are achieved using** more advanced LLMs such as **Gemini-1.5 Pro and GPT-4o**.
>
> To ensure a fair comparison, we have conducted additional experiments using different LLMs as backbones.
>
> **Baseline collection**
>
> For a thorough evaluation of SGU-SQL's performance, we added top-performing models from the BIRD leaderboard as baselines since the SPIDER leaderboard has remained static since February 2024. Specifically, we compared SGU-SQL against two categories of methods: (i) **Open-source models** with available paper and codes:  **MAC-SQL**, **Super-SQL**, **E-SQ**L and **CHESS**; and (ii) **Undisclosed methods** that have demonstrated strong performance: **PURPLE**, **Distillery** and **CHASE-SQL**.
>
> Table 1: Performance comparison on BIRD dev with different LLMs as backbones.
>
> | Execution Accuracy | MAC-SQL | PURPLE | E-SQL | CHESS | Distillery | CHASE-SQL | SGU-SQL (Ours) |
> | --- | --- | --- | --- | --- | --- | --- | --- |
> | GPT-4 | 59.59 | 60.71 | 58.95 | 61.37 | - | - | 61.80 |
> | GPT-4o | 65.05 | 68.12 | 65.58 | 68.31 | 67.21 | - | 69.28 |
> | Gemini-1.5 Pro | - | - | - | - | - | 73.14 | - |
>
> *Note that PURPLE, Distillery, and CHASE-SQL are closed-source models. We will update their results on GPT-4 and GPT-4o once their implementations become publicly available.*
>
> **Performance analysis**
> As shown in Table 1, our SGU-SQL achieves competitive performance across different LLM backbones. Specifically, we have the following observations:
>
> - Using **GPT-4** as the backbone, **SGU-SQL achieves the best performance** compared to other models using the same backbone.
> - With **GPT-4o**, SGU-SQL achieves 69.28% in terms of execution accuracy, **outperforming several strong baselines**: PURPLE (68.12%), CHESS (68.31%), E-SQL (65.58%) and Distillery (67.21%).
> - The only model showing higher performance is CHASE-SQL (released in October 2024), which uses Gemini 1.5 Pro as its backbone. Notably, **CHASE-SQL** incorporates a query fixer module that **leverages database execution feedback to guide LLMs to iteratively refine generated queries**. In contrast, **our model generates SQL queries in a single pass** without utilizing any execution feedback.
>
> **Manuscript Revision**:  We added the additional experiments with different LLMs as backbones in **Section C.3** (line994-1017).
>
> > **Regarding W2-b: Performance on SPIDER:** "Table 1 Spider. Many of the baseline methods are not Text-to-SQL specific methods, therefore performance is not competitive. additionally only show method is good on Spider is not enough. ideally test performance on test split by submitting to leaderboard for generation ability."
> >
>
> A:   We really appreciate your suggestion to further compare our model with more text-to-SQL models. In Table 1 of the original submission, we evaluated SGU-SQL on SPIDER with 11 text-to-SQL methods from 2 categories:
>
> - **Traditional structure-based methods**: RAT-SQL, LGESQL, RESDSQL, Graphix-T5
> - **LLM-based models**: DTS-SQL, CodeS, C³-SQL, Super-SQL, EPI-SQL, DIN-SQL, DAIL-SQL
>
> To provide a more comprehensive comparison, we have added top-performing models from the BIRD leaderboard as new baselines, including PURPLE, MAC-SQL, CHESS and CHASE-SQL. As shown in Table 2,  SGU-SQL consistently achieves superior performance compared to the strongest baselines on SPIDER in terms of Execution Accuracy. While the Spider leaderboard is no longer actively maintained, we'll further validate our model's performance by submitting results to the BIRD leaderboard.
>
> Table 2: Performance comparison between our proposed SGU-SQL and the strongest baselines on SPIDER in terms of Execution Accuracy.
>
> | Execution Accuracy | DAIL-SQL | DIN-SQL | PURPLE | Super-SQL  | MAC-SQL | CHESS | CHASE-SQL | SGU-SQL |
> | --- | --- | --- | --- | --- | --- | --- | --- | --- |
> | Backbone LLM | + GPT-4 | + GPT-4 | + GPT-4 | + GPT-4 | + GPT-4 | + GPT-4 | + Gemini-1.5 Pro | + GPT-4 |
> | Performance | 83.08 | 82.79 | 86.70 | 86.82 | 86.35 | 87.14 | 87.60 | 87.95 |
>
> **Manuscript Revision**: We added the results to Tables 1, 2 and 5 of our updated manuscript.

---

> > ### Author Response · Authors · 2024-11-24
> > **Response to Reviewer 53mq (3/3)**
> >
> > > **Regarding W3-a,b,c,d: Lack of ablation study.** "Need ablation study on effectiveness of graph representation of query (sec 3.1). Need ablation study on effectiveness of graph representation of database, compared with code representation. Need convincing ablation qualitative and qualitative results on the effectiveness of ‘structure-aware linking’. Need convincing ablation study to show “syntax-based decomposition” is effective."
> > >
> >
> > A: We appreciate the reviewer's suggestion for comprehensive ablation studies. Following your suggestion, we have conducted detailed experiments to validate the effectiveness of each component in SGU-SQL.
> >
> > Table 3: Ablation study on different components of SGU-SQL.
> >
> > | Execution Accuracy | SPIDER-dev | BIRD-dev |
> > | --- | --- | --- |
> > | Full Model | 87.95 | 61.80 |
> > | w/o query graph | 84.50 (-3.45) | 58.93 (-2.87) |
> > | w/o database graph | 85.81 (-2.14) | 58.26 (-3.54) |
> > | w/o structure-aware linking | 82.62 (-5.33) | 55.31 (-6.49) |
> > | w/o decomposition | 82.35 (-5.60) | 53.78 (-8.02) |
> >
> > As shown in Table 3, we have the following observations:
> >
> > - **Query graph**: Removing the query graph representation leads to significant performance drops (-3.45% on Spider-dev, -2.87% on BIRD-dev), demonstrating that our proposed query graph is crucial for understanding the intent behind the query.
> > - **Database Graph**: The ablation of the database graph results in performance decreases of -2.14% on Spider-dev and -3.54% on BIRD-dev.
> > - **Structure-aware Linking:** When removing structure-aware linking, we observe substantial performance degradation (-5.33% on Spider-dev, -6.49% on BIRD-dev). The more significant drop on BIRD emphasizes that our linking mechanism is particularly crucial for complex queries and databases, effectively bridging the semantic gap between natural language and database components while maintaining structural integrity.
> > - **Syntax-based Decomposition**: The ablation of our decomposition strategy leads to the most significant performance decrease (-5.60% on Spider-dev, -8.02% on BIRD-dev). These results validate our approach of breaking down complex queries into manageable components while preserving structural relationships.
> >
> > To summarize, each component contributes substantially to the overall performance, while the components are particularly crucial for handling complex queries (larger drops on BIRD).
> >
> > **Manuscript Revision**: To illustrate how each component contributes to accurate SQL generation, we added the detailed ablation study and analysis in Sectiion C (line947-1017) of our updated manuscript.
> >
> > > **Regarding W3-d: Comparison with advanced decomposition approach.** "Table 3 compares with other basic decomposition approach. relatively simple prompt decomposition method can get on par text-to-sql performance of MAC-SQL https://arxiv.org/pdf/2312.11242 (86.75 vs 87.95).”
> > >
> >
> > A: Thanks for your careful review and constructive suggestions. To provide a comprehensive comparison, we included state-of-the-art decomposition-based text-to-SQL models, including DIN-SQL, ACT-SQL, MAC-SQL, in our evaluation. These models leverage advanced decomposition strategies to break down complex SQL queries into simpler sub-queries or subtasks, facilitating more accurate and efficient SQL generation.
> >
> > **Model selection:**
> > Following your suggestion, we conducted additional experiments to compare our SGU-SQL with other advanced decomposition-based methods, including:
> >
> > - **DIN-SQL** breaks down the Text-to-SQL task into 4 fixed sub-problems (schema linking, classification, SQL generation, and self-correction).
> > - **ACT-SQL** proposes an in-context learning method utilizing automatically-generated Chain-of-Thought (Auto-CoT) for decomposition.
> > - **MAC-SQL** uses LLM as an Agent of Decomposer to break complex questions into simpler sub-questions.
> >
> > **Performance analysis:**
> >
> > **Table 4** below presents the performance of **SGU-SQL** compared to the aforementioned decomposition-based approaches on the Spider dataset, measured by **Execution Accuracy**.
> >
> > Table 4: Performance comparison between SGU-SQL and advanced decomposition methods.
> >
> > | Execution Accuracy(EX) | DIN-SQL+GPT-4 | ACT-SQL+GPT-4 | MAC-SQL+GPT-4 | SGU-SQL+GPT-4 |
> > | --- | --- | --- | --- | --- |
> > | SIPDER-dev | 82.79 | 82.90 | 86.35(reproduced) | 87.95 |
> >
> > Our SGU-SQL achieves 87.95% execution accuracy on SPIDER-dev, outperforming all these methods. The key distinction of our approach is that it dynamically decomposes queries based on their syntax structure, rather than either using fixed decomposition patterns (like DIN-SQL) or purely relying on LLM's black-box understanding (like ACT-SQL, MAC-SQL). This syntax-aware decomposition strategy proves more effective for handling complex SQL generation tasks.
> >
> > **Manuscript Revisions:** We have added the results in Table 5 to the updated manuscript, and the analysis can be found in Section C.2 (line972-992) of our revision.

---

> > ### Comment · Reviewer_53mq · 2024-11-26
> > **Thanks for the response**
> >
> > Thank you to the authors for providing detailed ablation studies.
> >
> > However, my primary concern remains that the method's performance on BIRD is significantly lower (61 vs. SOTA at 74), a gap of 13%. In contrast, the model achieves SOTA on SPIDER. This disparity suggests the method may be heavily overfitted to SPIDER.
> >
> > I also find it unlikely that the use of a weaker foundation model (GPT-4 in this method vs. GPT-4-o in others) would result in a performance drop of 13% on BIRD. The significant drop in performance indicates some fundamental components may be missing in the method.

---

> ### Author Response · Authors · 2024-11-26
> **Second-Round Response to Reviewer 53mq (1/2)**
>
> Dear Reviewer 53mq,
>
> Thanks a lot for your timely response and careful review. We are delighted that you found the new ablation studies are helpful. We also greatly appreciate the opportunity to further clarify the remaining concerns.
>
> ### **Clarification**
>
> To avoid any potential confusion, we first offer the following clarification:
>
> - The SOTA model (the only model outperforming ours) is **CHASE-SQL+Gemini-1.5 Pro** $\textcolor{maroon}{(73.14\\%)}$. Google Research implemented this model using **Gemini 1.5 Pro as its backbone**, and they just released this paper in October 2024, with the **code still unavailable.**
> - **SGU-SQL+GPT-4o** achieves $\textcolor{maroon}{69.28\\%}$, outperforming the other strong baselines: E-SQL+GPT-4o (65.58%), Distillery+GPT-4o (67.21%), PURPLE+GPT-4o (68.12%) and CHESS+GPT-4o (68.31%).
>
> We provide detailed explanations and analysis as follows:
>
> > **Regarding the performance on BIRD**: ”However, my primary concern remains that the method's performance on BIRD is significantly lower (61 vs. SOTA at 74), a gap of 13%.“
> >
>
> Regarding your concern about the performance on BIRD, we have provided some related experiments and analysis in ***Table 1 of the previous response.***
>
> Table 1: Performance comparison on BIRD dev with different LLMs as backbones.
>
> | Backbone LLM | MAC-SQL | PURPLE | E-SQL | CHESS | Distillery | CHASE-SQL | SGU-SQL (Ours) |
> | --- | --- | --- | --- | --- | --- | --- | --- |
> | **+GPT-4** | 59.59 | 60.71 | 58.95 | 61.37 | - | - | **61.80** |
> | **+GPT-4o** | 65.05 | 68.12 | 65.58 | 68.31 | 67.21 | - | **69.28** |
> | **+Gemini-1.5 Pro** | - | - | - | - | - | **73.14** | - |
>
> To make it clearer, we reorganized these results in the following table to compare their overall performance.
>
> Table 2: Performance comparison on BIRD dev.
>
> | Execution Accuracy | MAC-SQL + GPT-4o | PURPLE + GPT-4o | E-SQL + GPT-4o | CHESS + GPT-4o | Distillery + GPT-4o | SGU-SQL + GPT-4o | CHASE-SQL+ Gemini-1.5 Pro |
> | --- | --- | --- | --- | --- | --- | --- | --- |
> | BIRD-dev | 65.05 | 68.12 | 65.58 | 68.31 | 67.21 | **69.28** | **73.14** |
>
> As shown in Tables 1 and 2, we have the following observations:
>
> - The SOTA model (the only model outperforming ours) is **CHASE-SQL+Gemini-1.5 Pro** $\textcolor{maroon}{(73.14\\%)}$. Google Research implemented this model using **Gemini 1.5 Pro**, and they just released the paper in October 2024, with the **code still unavailable.**
> - **SGU-SQL+GPT-4o** achieves $\textcolor{maroon}{69.28\\%}$, outperforming the strong baselines: E-SQL+GPT-4o (65.58%), Distillery+GPT-4o (67.21%), PURPLE+GPT-4o (68.12%) and CHESS+GPT-4o (68.31%). (We didn’t compare our model with CHASE-SQL+GPT-4o since CHASE-SQL is still closed-source and unable to integrate other LLMs.)
> - **SGU-SQL+GPT-4** achieves the best performance compared to the other baselines using GPT-4 as the backbone.
>
> > **Regarding the concern of possible overfitting on Spider**: “In contrast, the model achieves SOTA on SPIDER. This disparity suggests the method may be heavily overfitted to SPIDER.”
> >
>
> As shown in Table 3, all models show a similar performance pattern across both datasets. This consistent pattern suggests that the **variations in performance stem from BIRD's inherent complexity** rather than model overfitting. (BIRD is currently the largest text-to-SQL dataset. It comprises over 12,751 question-SQL pairs, spanning 95 large databases with a total size of 33.4 GB.)
>
> Table 3: Performance comparison in terms of Execution Accuracy.
>
> | Execution Accuracy | DAIL-SQL + GPT-4 | DIN-SQL + GPT-4 | PURPLE + GPT-4 | Super-SQL + GPT-4 | MAC-SQL + GPT-4 | CHESS + GPT-4 | SGU-SQL + GPT-4 |
> | --- | --- | --- | --- | --- | --- | --- | --- |
> | BIRD-dev | 54.34 | 50.72 | 60.71 | 58.60 | 57.59 | 61.37 | 61.80 |
> | Spider-dev | 83.08 | 82.79 | 86.70 | 86.82 | 86.35 | 87.14 | 87.95 |
>
> > **Regarding the performance gap of using different LLMs**: “I also find it unlikely that the use of a weaker model (GPT-4 in this method vs. GPT-4o in others) would result in a performance drop of 13% on BIRD. The significant drop in performance indicates some fundamental components may be missing in the method.”
> >
>
> Table 4: Performance gap of using different LLMs.
> | Model | GPT-4 | GPT-4o | Improvement |
> | --- | --- | --- | --- |
> | E-SQL | 58.95 | 65.58 | +6.63% |
> | MAC-SQL | 59.59 | 65.05 | +5.46% |
> | PURPLE | 60.71 | 68.12 | +7.41% |
> | CHESS | 61.37 | 68.31 | +6.94% |
> | SGU-SQL | 61.80 | 69.28 | +7.48% |
>
> Looking at the results across all models, switching from GPT-4 to GPT-4o typically brings a **5-8%** improvement:
>
> - All models show similar improvements when switching to GPT-4o.
> - Our model's improvement (+7.48%) aligns with this pattern.
>
> Thanks again for your active response and professional comments. Your expertise and guidance have been invaluable in shaping our revisions. Should there be any remaining concerns, we would be more than happy to provide additional information or explanations to assist in your review process.

---

> ### Author Response · Authors · 2024-12-02
> **Second-Round Response to Reviewer 53mq (2/2)**
>
> Dear Reviewer 53mq,
>
> Thank you for your expertise and insightful comments. To achieve a thorough evaluation of SGU-SQL, we have conducted additional experiments to compare our model with CHASE-SQL by using more advanced LLMs as backbones during the past few days,
>
> **Model Selection**
> * **Gemini 1.5 Pro**: When conducting experiments with Gemini 1.5 Pro, we noted that there are currently 5 available versions: `gemini-1.5-pro-latest`, `gemini-1.5-pro`, `gemini-1.5-pro-001`, `gemini-1.5-pro-002` and `gemini-1.5-pro-exp-0827`. Since CHASE-SQL did not specify its model version, we conducted experiments with two versions - `gemini-1.5-pro` and `gemini-1.5-pro-latest` - for a comprehensive comparison.
> * **Claude 3.5 Sonnet**: Additionally, as CHASE-SQL reported performance with Claude 3.5 Sonnet, we also included it as a backbone model.
>
> **Performance Analysis**
>
> Table 1: Performance on BIRD-dev in terms of Execution Accuracy.
>
> | Model | Gemini 1.5 Pro | Claude 3.5 Sonnet |
> |-------|----------------|-------------------|
> | CHASE-SQL | 73.01% (`unknown version`) | 69.53% |
> | SGU-SQL | 72.76% (`gemini-1.5-pro`) 72.93% (`gemini-1.5-pro-exp-0827`) | 70.36% |
>
> *Due to time and API budget limits, we have currently only evaluated our model's performance with Gemimi 1.5 Pro and Claude 3.5. We plan to conduct more comprehensive experiments with other baselines using these advanced LLMs in future work.*
>
> The experiment results demonstrate several key insights:
>
> * First, when using **Gemini 1.5 Pro** as the backbone, SGU-SQL achieves highly competitive results (72.76% with gemini-1.5-pro and 72.93% with gemini-1.5-pro-exp-0827) compared to CHASE-SQL (73.01%).
> * Furthermore, with **Claude 3.5 Sonnet** as the backbone, SGU-SQL (70.36%) slightly outperforms CHASE-SQL (69.53%). This improvement suggests that our method may better leverage Claude's capabilities through its structured decomposition approach.
> * Our approach demonstrates robust and competitive performance across different state-of-the-art language models.
>
>
> Thanks again for your expertise and professional comments. Your suggestions and guidance have been invaluable in shaping our revisions. As the deadline for the Rebuttal approaches, we want to reach out and inquire whether our responses have addressed your concerns. If there are any remaining concerns or areas that require further clarification, we would greatly appreciate it if you could provide us with additional feedback.
>
> Best,
>
> Authors of Paper7028

---

> ### Author Response · Authors · 2024-12-02
> **Kindly Requesting Your Feedback on Our Response**
>
> Dear Reviewer 53mq,
>
> Thanks again for taking the time to review our paper and providing valuable feedback. We have carefully responded to your latest concerns, conducting extensive experiments comparing our approach with CHASE-SQL on BIRD. We have also provided detailed analysis by using different advanced LLMs as backbones. We hope these analyses can address your concerns.
>
> As the author-reviewer discussion deadline approaches, we would be very grateful if you could share your valuable feedback. We are committed to strengthening our work and would be grateful for your further insights.
>
> Best regards,
>
> Authors of Paper7028

---

> > ### Comment · Reviewer_53mq · 2024-12-03
> > **Thank you for the rebuttal.**
> >
> > Thanks for the experiments. The increased performance of the proposed method mitigate my concerns. I increased the scores.

---

> ### Author Response · Authors · 2024-12-03
> **Thanks for Your Recognition of Our Work**
>
> Dear Reviewer 53mq,
>
> We would like to deeply express our gratitude for your recognition of our efforts to address your concerns. Your expertise and constructive feedback really help us advance our research. We will add the additional experiments and discussion in our final revision.
>
> Best regards,
>
> Authors of Paper7028

---

### Author Response · Authors · 2024-12-01
**General Response to Area Chairs and All Reviewers (1/2)**

Dear Area Chairs and Reviewers,

We sincerely thank you for your time, effort and invaluable feedback during the author-reviewer discussion phase. Your insights have been crucial in helping us refine and improve our work. To facilitate the next stage of review and discussion, we would like to summarize the key contributions of our paper, the main points raised during the discussion and our response.

### **Motivation**

LLM-based text-to-SQL models often struggle to comprehend complex database structures and accurately interpret user intentions. Decomposition-based methods have been proposed to enhance the performance of LLMs on complex tasks, but **decomposing SQL generation into subtasks is non-trivial** due to the declarative structure of SQL syntax and the intricate connections between query concepts and database elements. In this paper, we propose a novel **structure-guided text-to-SQL framework** (SGU-SQL) that incorporates **syntax-based prompting** to enhance the SQL generation capabilities of LLMs.


### **Contribution**

SGU-SQL offers significant improvements over existing approaches in two key aspects.

- **Syntax-based Prompting.** While traditional methods attempt to generate entire SQL queries in one step or rely on simple decomposition strategies, SGU-SQL breaks down the complex generation task in a syntax-aware manner. This ensures that the generated queries maintain both semantic accuracy (correctly capturing user intentions) and syntactic correctness (following proper SQL structure).
- **Combine with Structure-aware Linking.** Our framework's unique combination of structure-aware linking and syntax-guided decomposition creates a robust bridge between user intentions and database structures. The structure-aware linking ensures that each natural language concept is correctly mapped to the corresponding database elements, while the syntax-guided decomposition ensures these mappings are properly utilized in constructing the SQL query.

### **Main discussion points and our responses.**

- **Impact of Structure Linking**. The structure-aware linking creates a foundation for accurate decomposition by establishing precise mappings between natural language elements and database schema components. Following the suggestions from Reviewer 53mq, b8Gr and vRvu, we added more details about the **design principles**, **training process** and **efficiency analysis** of the structure linking in Section 3.1.3, and conducted a comprehensive ablation study on the backbone GNN models for structure learning (Appendix C.1 and D.2).
- **Performance on BIRD and SPIDER**. Following the suggestions from Reviewer 53mq and EsJa,  we included the **latest baselines** and conducted additional experiments with **different LLMs as backbones** to achieve a thorough evaluation of SGU-SQL in (Appendix C.3).
- **Comparison with CoT and Few-shot-based Methods**. Following the suggestions from Reviewer 53mq and b8Gr, we conducted additional experiments to compare our SGU-SQL with other advanced prompting strategies to verify the effectiveness of our syntax-based decomposition strategy (Appendix C.2).
- **Additional Experiment on SPIDER-2.** SPIDER-2 is currently the most challenging text-to-SQL dataset available. Following the suggestion from Reviewer vRvu, we extended our evaluation on **Spider 2.0-Snow** and **Spider 2.0-Lite** (Appendix D.1).
- **Future Work.** Discussing potential extensions is crucial for the research community. Following the suggestion from Reviewer vRvu, we have identified several promising future research directions from **three perspectives** (Appendix E).

### **Detailed analysis of concerns mentioned by Reviewer EsJa**

After the first round of responses, Reviewers 53mq and EsJa raised insightful questions regarding our model's performance on the BIRD and Spider benchmarks. In our second-round response, we provided thorough clarifications and additional analyses to address their concerns. **Reviewer 53mq** has acknowledged our explanations and positively updated their evaluation, while we have not yet received feedback from **Reviewer EsJa**. To prevent any potential confusion in subsequent reviews, we now present a detailed examination of these concerns.

**Performance on BIRD**

For a thorough evaluation of SGU-SQL's performance, we included top-performing models from the BIRD leaderboard as baselines.

**a. Baseline selection**

As shown in Table 1, the current BIRD leaderboard features a mix of academic research and industrial solutions. For a comprehensive evaluation, we selected baselines from two categories:

- **Open-source models** with published papers and available code (MAC-SQL, Super-SQL, E-SQL, **CHESS**).
- **High-performing methods** with only available papers (**PURPLE**, **Distillery**, **CHASE-SQL**).

---

> ### Author Response · Authors · 2024-12-04
> **General Response to Area Chairs and All Reviewers (2/2)**
>
> Table 1: Top-performing text-to-SQL models ranked by BIRD leaderboard.
>
> | Rank | Model | Backbone LLM | BIRD-dev | Release Date | Code Status | Organization | Type |
> | --- | --- | --- | --- | --- | --- | --- | --- |
> | 1 | **CHASE-SQL** | Gemini | 73.14* | 2 Oct 2024 | Not Available | Google Cloud | Industry |
> | 2 | DSAIR | GPT-4o | 74.32 | No Research Paper | Not Available | AT&T | Industry |
> | 3 | ExSL | granite-34b-code | 72.43 | No Research Paper | Not Available | IBM Research | Industry |
> | 4 | OpenSearch-SQL | GPT-4o | 69.30 | No Research Paper | Lots of Bugs | Alibaba Cloud | Industry |
> | 5 | **Distillery** | GPT-4o | 67.21 | 14 Aug 2024 | Not Available | Distyl AI Research | Industry |
> | 6 | **CHESS** | GPT-4o | 68.31 | 27 May 2024 | Available | Stanford University | Academic |
> | 7 | Insights AI | UNK | 72.16 | No Research Paper | Not Available | Uber Freight | Industry |
> | 8 | **PURPLE** | GPT-4o | 68.12 | 29 Mar 2024 | Not Available | Fudan University | Academic |
>
> **Results reported in the paper.*
>
> To summarize, from the top-8 methods in the BIRD leaderboard, we included **PURPLE (8th)**, **CHESS (6th)**, **Distillery (5th)**, and **CHASE-SQL (1st)** in our comparisons. We excluded the remaining 4 methods (DSAIR, ExSL, OpenSearch-SQL, Insights AI) since they are all industrial solutions **without any released instructions** (papers and technical reports) or **accessible code**.
>
> **b. Performance analysis**
>
> Table 2: Performance comparison on BIRD dev with different LLMs as backbones.
>
> | Backbone LLM | MAC-SQL | PURPLE | E-SQL | CHESS | Distillery | CHASE-SQL | SGU-SQL (Ours) |
> | --- | --- | --- | --- | --- | --- | --- | --- |
> | **+GPT-4** | 59.59 | 60.71 | 58.95 | 61.37 | - | - | **61.80** |
> | **+GPT-4o** | 65.05 | 68.12 | 65.58 | 68.31 | 67.21 | - | **69.28** |
> | **+Gemini-1.5 Pro** | - | - | - | - | - | 73.14 | **72.93** |
> | **+Claude 3.5 Sonnet** | - | - | - | - | - | 69.53 | **70.36** |
>
> *Due to time and API budget limits, we have currently only evaluated our model's performance with Gemimi 1.5 Pro and Claude 3.5. We plan to conduct more comprehensive experiments with other baselines using these advanced LLMs in future work.*
>
> As shown in Table 2, we have the following observations:
>
> - **SGU-SQL+GPT-4** achieves the best performance compared to the other baselines using GPT-4 as the backbone.
> - **SGU-SQL+GPT-4o** achieves 69.28%, outperforming the strong baselines: E-SQL+GPT-4o (65.58%), Distillery+GPT-4o (67.21%), PURPLE+GPT-4o (68.12%) and CHESS+GPT-4o (68.31%). (We didn’t compare our model with CHASE-SQL+GPT-4o since CHASE-SQL is still closed-source and unable to integrate other LLMs.)
> - When using **Gemini 1.5 Pro** as the backbone, SGU-SQL achieves highly competitive results (72.76% with `gemini-1.5-pro` and 72.93% with `gemini-1.5-pro-exp-0827`) compared to CHASE-SQL (73.01%).
> - With **Claude 3.5 Sonnet** as the backbone, SGU-SQL (70.36%) slightly outperforms CHASE-SQL (69.53%). This improvement suggests that our method may better leverage Claude's capabilities through its structured decomposition approach.
>
> To summarize, our approach demonstrates robust and competitive performance across different state-of-the-art language models, which verifies the effectiveness of our proposed syntax-based prompting strategy.
>
> **Performance on Spider**
>
> **a. Baseline selection**
>
> The SPIDER leaderboard has remained static since February 2024, with only a select number of recent papers reporting their performance on this benchmark. Our comparison encompasses a comprehensive range of methods, including:
>
> - **Top-performing models in Spider leaderboard**：DAIL-SQL (2th, 3th), DIN-SQL (4th), C3-SQL (6th), RESDSQL(8th).
> - **Recent innovations**: DTS-SQL, CodeS, Super-SQL, EPI-SQL, PURPLE, MAC-SQL, CHESS and CHASE-SQL.
>
> Note that **Miniseek** (1st) is the top-performing model on the Spider leaderboard, which achieves an execution accuracy of 91.2%. We didn't include this model since it is still anonymous, with **both the paper and code unavailable**.
>
> **b. Performance analysis**
>
> Table 3: Performance comparison on SPIDER in terms of Execution Accuracy.
>
> | Execution Accuracy | DAIL-SQL | DIN-SQL | PURPLE | Super-SQL | MAC-SQL | CHESS | CHASE-SQL | SGU-SQL |
> | --- | --- | --- | --- | --- | --- | --- | --- | --- |
> | Backbone LLM | + GPT-4 | + GPT-4 | + GPT-4 | + GPT-4 | + GPT-4 | + GPT-4 | + Gemini-1.5 Pro | + GPT-4 |
> | Performance | 83.08 | 82.79 | 86.70 | 86.82 | 86.35 | 87.14 | 87.60 | 87.95 |
>
> As shown in Table 3,  SGU-SQL consistently achieves superior performance compared to the strongest baselines on SPIDER in terms of Execution Accuracy.
>
>
> Thank you again for your time and expertise throughout this review process. We deeply appreciate your expertise and recognition of our work. Each point raised has given us new perspectives to consider. **Hope that this summary could facilitate the next stage of review and discussion**.
>
> Best regards,
>
> Authors of Paper7028

---

### Meta-Review · Area_Chair_Kic9 · 2024-12-17

**Metareview:**

This paper presents SGU-SQL, a model that leverages syntactic and structural information of user queries and database schemas to generate more accurate SQL queries. There are three components in the system. Given a user query and a database schema, a graph encoder matches phrases in the user query to corresponding database elements (entity/schema linking), such linking information is then used to decompose the original task into sub-tasks. Those sub-tasks are subsequently solved by LLMs by predicting their corresponding SQL queries. The authors provided extensive experiment results, claiming that incorporating structural information can effectively improve text-to-SQL accuracy. However, due to issues with writing it is unclear how the method actually works (more details below).

**Strengths:**

* While the idea of leveraging syntactic and structural information for text-to-SQL is not new (53mq), the idea of composing a graph-encoding approach together with LLMs is interesting and effective (53mq, vRvu, EsJa), and could help “reduce the input of LLM(s) to some extent” (b8Gr)

* Evaluation is fairly comprehensive (EsJa), demonstrating “strong results on the most popular SQL datasets SPIDER and BIRD” (vRvu), “achieving substantial improvements over naive few-shot prompting and chain-of-thought (CoT) methods” (EsJa).

**Weaknesses:**

The following issues are addressed during the rebuttal phase:

* Performance is not competitive on more complex datasets like BIRD (53mq) — authors provided more through evaluation results that shows strong results on BIRD with more recent LLMs.

* Ablation needed to show impact of different components (53mq) — addressed with additional ablation results

* Evaluation on more complex text-to-sql tasks like SPIDER-2 (vRvu) — partially addressed with preliminary results on SPIDER-2.
Comparison with more recent SoTA methods like  CHESS, Distillery, and CHASE-SQL — addressed by including results from these models, where the proposed SGU-SQL still outperforms these models.

The major issue yet to be addressed is clarity and technical writing:

* 53mq: “Method writing should be clearer”

* vRvu: “Specific discussion on the model used for structure linking seem insufficient”

* EsJa: “crucial tables … have been relegated to the appendix. These tables contain valuable data and should be included in the main body of the paper”

After carefully reading the submission, I agree with the reviewers comments on writing issues. For example, it is not clear how the model was trained, and how it works end-to-end at inference time. Note that the proposed approach has two components: a trainable GNN that learns structural information of natural language queries and DB schemas, and a prompting method using LLMs to generate SQL queries. It is not clear how the trainable component works with the LLM. While the authors provided further clarifications in the updated version of the draft, without a running example to illustrate the entire approach, it is still difficult for readers to understand how the approach work end-to-end.

Other missing technical information includes the models / methods used to generate the syntactic and semantic parse of a natural language query. Indeed, to my knowledge, the relation types in Table 9 do not map to any existing well-known syntactic / semantic taxonomy.

Therefore, given the remaining issues with technical presentation. I believe that the paper could indeed benefit from another round of major revision. Therefore, the recommendation is Rejection.

**Additional Comments On Reviewer Discussion:**

See above

---

### Decision · Program_Chairs · 2025-01-22

Reject